# A comprehensive and synthetic dataset for global, regional and national greenhouse gas emissions by sector 1970-2018 with an extension to 2019

Jan C. Minx[1,2,*], William F. Lamb[1,2,*], Robbie M Andrew[3], Josep G. Canadell[4], Monica Crippa[5], Niklas Döbbeling[1], Piers M. Forster[2], Diego Guizzardi[5], Jos Olivier[6], Glen P. Peters[3], Julia Pongratz[7,8], Andy Reisinger[9], Matthew Rigby[10], Marielle Saunois[11], Steven J. Smith[12], Efisio Solazzo[5], Hanqin Tian[13]

[1] Mercator Research Institute on Global Commons and Climate Change, Berlin, 10827, Germany
[2] Priestley International Centre for Climate, School of Earth and Environment, University of Leeds, Leeds; LS2 9JT, UK
[3] CICERO Center for International Climate Research, Oslo, 0318 Norway
[4] Global Carbon Project, CSIRO Oceans and Atmosphere, Canberra, Australia
[5] European Commission, Joint Research Centre, Ispra, Italy
[6] PBL Netherlands Environmental Assessment Bureau, the Hague, the Netherlands
[7] Ludwig-Maximilians-Universität München, Luisenstr. 37, 80333 Munich, Germany
[8] Max Planck Institute for Meteorology, Bundesstrasse 53, 20146 Hamburg, Germany
[9] Institute for Climate, Energy and Disaster Solutions, Fenner School of Society & Environment, Building 141, Linnaeus Way, The Australian National University, Canberra, ACT, 2600, Australia
[10] School of Chemistry, University of Bristol, Bristol, BS8 1TS, UK
[11] Laboratoire des Sciences du Climat et de l'Environnement, LSCE-IPSL (CEA-CNRS-UVSQ), Université Paris-Saclay 91191 Gif-sur-Yvette, France
[12] Joint Global Change Research Institute, Pacific Northwest National Laboratory, College Park, MD 20740 USA
[13] International Center for Climate and Global Change Research, School of Forestry and Wildlife Sciences, Auburn University, Auburn, AL 36849, USA
[*] *Correspondence to*: Jan C. Minx (minx@mcc-berlin.net) and William F. Lamb (lamb@mcc-berlin.net)

## Abstract

To track progress towards keeping global warming well below 2°C or even 1.5°C, as agreed in the Paris Agreement, comprehensive up-to-date and reliable information on anthropogenic emissions and removals of greenhouse gas emissions (GHG) is required. Here we compile a new synthetic dataset on anthropogenic GHG emissions for 1970-2018 with a fast-track extension to 2019. Our dataset is global in coverage and includes $CO_2$ emissions, $CH_4$ emissions, $N_2O$ emissions as well as those from fluorinated gases (F-gases: HFCs, PFCs, $SF_6$, $NF_3$) and provides country and sector detail. We build this dataset from the version 6 release of the "Emissions Database for Global Atmospheric Research" (EDGAR v6) and three bookkeeping models for $CO_2$ emissions from land use, land-use change and forestry (LULUCF). We assess the uncertainties of global greenhouse gases at the 90% confidence interval (5th-95th percentile range) by combining statistical analysis and comparisons of global emissions inventories and top-down atmospheric measurements with an expert judgement informed by the relevant scientific literature. We identify important data gaps for F-gas emissions. The agreement between EDGAR and atmospheric-based emissions estimates is relatively close for some F-gas species (~10% or less) but the estimates can differ by an order of magnitude or more for others. When aggregated, the EDGAR v6 F-gas total agrees with top-down estimates to within around

10% in recent years. However, emissions from excluded F-gas species such as chlorofluorocarbons (CFCs) or hydrochlorofluorocarbons (HCFCs) are cumulatively larger than the sum of the reported species. Using global warming potential values with a 100 year time horizon from the Sixth Assessment Report by the Intergovernmental Panel on Climate Change (IPCC), global GHG emissions in 2018 amounted to $58\pm6.1$ GtCO$_2$eq consisting of: CO$_2$-FFI $38\pm3.0$ GtCO$_2$, CO$_2$-LULUCF $5.7\pm4.0$ GtCO$_2$, CH$_4$ $10\pm3.1$ GtCO$_2$eq, N$_2$O $2.5\pm1.5$ GtCO$_2$eq and F-gases $1.3\pm0.40$ GtCO$_2$eq. Initial estimates suggest further growth in GHG emissions by 1.3 GtCO$_2$eq to reach $59\pm6.6$ GtCO$_2$eq in 2019. Our analysis of global trends in anthropogenic GHG emissions over the past five decades (1970-2018) highlights a pattern of varied, but sustained emissions growth. There is high confidence that global anthropogenic GHG emissions have increased every decade and emissions growth has been persistent across different (groups of) gases. There is also high confidence that global anthropogenic GHG emission levels were higher in 2009-2018 than in any previous decade and GHG emission levels grew throughout the most recent decade. While the average annual GHG emissions growth rate slowed between 2009-2018 (1.2% yr$^{-1}$) compared to 2000-2009 (2.5% yr$^{-1}$), the absolute increase in average decadal GHG emissions from the 2000s to the 2010s was the largest since the 1970s, and within all human history. Our analysis further reveals that there are no global sectors that show sustained reductions in GHG emissions. There are a number of countries that have reduced GHG emissions over the past decade, but these reductions are comparatively modest and outgrown by much larger emissions growth in some developing countries such as China, India and Indonesia. There is a need to further develop independent, robust, and timely emission estimates across all gases. As such, tracking progress in climate policy requires substantial investments in independent GHG emission accounting and monitoring, as well as in national and international statistical infrastructures. The data associated with this article (Minx et al., 2021) can be found at , https://doi.org/10.5281/zenodo.5053055.

**1 Introduction**

By signing the Paris Agreement, countries acknowledged the necessity to keep the most severe climate change risks in check by limiting warming to well below 2°C, and to pursue efforts to limit warming to 1.5°C (UNFCCC, 2015). This requires rapid and sustained greenhouse gas (GHG) emission reductions towards net zero carbon dioxide ($CO_2$) emissions well within the 21$^{st}$ century along with deep reductions in non-$CO_2$ emissions (Rogelj et al., 2015, 2018a). Transparent, comprehensive, consistent, accurate and up-to-date inventories of anthropogenic GHG emissions are crucial to track progress by countries, regions and sectors in moving towards these goals.

However, it is challenging to accurately track the recent GHG performance of countries and sectors. While there is a growing number of global emissions inventories, only a few of them provide a wide coverage of gases, sectors, activities, and countries or regions that are sufficiently up-to-date to comprehensively track progress and thereby aid discussions in science and policy. Table 1 provides an overview of global emission inventories. Many inventories focus on individual gases and subsets of activities. Few provide sectoral detail and particularly for non-$CO_2$ GHG emissions there is often a considerable time-lag in reporting. GHG emissions reporting under the United Nations Framework Convention on Climate Change (UNFCCC) provides reliable, comprehensive and up-to-date statistics for Annex I countries across all major GHGs. Non-Annex I countries – except least developed countries and small island state for which this is not mandatory – provide GHG emissions inventory information through biennial update reports (BURs), but with much less stringent reporting requirements in terms of sector, gas and time coverage (Deng et al., 2021; Gütschow et al., 2016). As a result, many still lack a well-developed statistical infrastructure to provide detailed reports (Janssens-Maenhout et al., 2019).

Here we describe a new, comprehensive and synthetic dataset for global, regional and national GHG emissions by sector for 1970-2018 with a fast-track extension to 2019. Our focus is on GHG emissions from anthropogenic activities only. We build the dataset from recent releases of the "Emissions Database for Global Atmospheric Research" version 6 (EDGARv6) for $CO_2$ emissions from fossil fuel combustion and industry (FFI), $CH_4$ emissions, $N_2O$ emissions, and fluorinated gases (F-gases: HFCs, PFCs, $SF_6$ and $NF_3$) (Crippa et al., 2021). For completeness we add net $CO_2$ emissions from land use, land-use change and forestry ($CO_2$-LULUCF) from three bookkeeping models (Gasser et al., 2020; Hansis et al., 2015; Houghton and Nassikas, 2017). We provide an assessment of the uncertainties in each GHG at the 90% confidence interval (5$^{th}$-95$^{th}$ percentile) by combining statistical analysis and comparisons of global emissions inventories with an expert judgement informed by the relevant scientific literature.

**Table 1 – Overview of global inventories of GHG emissions**

| Dataset Name | Short Name | Version | Gases | Geo-graphic coverage | Activity split | Time period | Reference | Link |
|---|---|---|---|---|---|---|---|---|
| Emissions Database for Global Atmospheric Research | EDGAR | 6.0 | $CO_2$-FFI, $CH_4$, $N_2O$, F-gases: HFCs, PFCs, $SF_6$, $NF_3$. | 228 countries; global | 4 main sectors, 24 subsectors | 1970-2018 | (Crippa et al., 2021) | https://edgar.jrc.ec.europa.eu/report_2021 |
| Potsdam Real-time Integrated Model for probabilistic Assessment of emissions Paths | PRIMAP-hist | 2.3.1 | $CO_2$-FFI, $CH_4$, $N_2O$, F-gases: HFCs, PFCs, $SF_6$, $NF_3$. | All UNFCCC member states, most non-UNFCCC territories | 4 sectors | 1750-2019 | Gütschow et al. (2016, 2021) | https://www.pik-potsdam.de/paris-reality-check/primap-hist/ |
| Community Emissions Data System | CEDS | v_2021_02_05 | $SO_2$, $NO_x$, BC, OC, $NH_3$, NMVOC, CO, $CO_2$, $CH_4$, $N_2O$ | 221 countries | 60 sectors | 1750-2019 (1970-2019 for CH4 and N2O) | Hoesly et al. (2018); McDuffie et al. (2020); O'Rourke et al. (2021) | http://www.globalchange.umd.edu/ceds/ |
| UNFCCC: Annex I Party GHG Inventory Submissions | | 2021 | $CO_2$, $CH_4$, $N_2O$, $NO_x$, CO, NMVOC, $SO_2$, F-gases: HFCs, PFCs, $SF_6$, $NF_3$ | Parties included in Annex I to the Convention | Energy, industry, agriculture, LULUCF, waste | 1990-2019 | | https://unfccc.int/ghg-inventories-annex-i-parties/2021 |
| GCP: Global Carbon Budget | GCP-GCB | 2020 | $CO_2$-FFI, $CO_2$-LULUCF | Global, 259 countries for FFI | 5 main sectors, 14 subsectors | $CO_2$-LULUCF: 1850-2019  $CO_2$-FFI: 1750-2019 | Friedlingstein et al. (2020) | https://doi.org/10.18160/GCP-2020 |
| Global, Regional, and National Fossil-Fuel $CO_2$ Emissions | CDIAC-FF | V2017 | $CO_2$-FFI | 259 countries, global | 5 main categories | 1751-2017 | Gilfillan et al. (2020) | https://energy.appstate.edu/research/work-areas/cdiac-appstate |
| Energy Information Administration International Energy Statistics | EIA | 2021 | $CO_2$-FFI | 230 countries, global | 3 fuel types | 1980-2018; 1949-2018 (global) | EIA (2021) | https://www.eia.gov/international/data/world |
| BP Statistical Review of World Energy | BP | 2021 70th edition | $CO_2$-FFI | 108 countries, 7 regions | 8 activities, 3 fossil and 3 other fuel types | 1965-2019 | BP (2021) | https://www.bp.com/en/global/corporate/energy-economics/statistical-review-of-world-energy.html |

| | | | | | | | | |
|---|---|---|---|---|---|---|---|---|
| International Energy Agency $CO_2$ Emissions from Fuel Combustion | IEA | 2021 | $CO_2$-FFI | 190 countries | 3 fossil fuels, 6 sectors | 1971-2020; OECD: 1960-2020 | IEA (2021a, 2021b) | https://webstore.iea.org/co2-emissions-from-fuel-combustion-2019-highlights |
| PKU-FUEL | | | $CO_2$, CO, $PM_{2.5}$, $PM_{10}$, TSP, BC, OC, $SO_2$, $NO_x$, $NH_3$, PAHs | Global (0.1 degree grid cells) | 6 sectors, 5 fuel types, | 1960-2014 | | http://inventory.pku.edu.cn/ |
| Carbon Monitor | | | $CO_2$-FFI | 11 countries, global | 6 sectors | 2019- | Liu et al. (2020) | https://carbonmonitor.org/ |
| Bookkeeping of land-use emissions | BLUE | 2020 | $CO_2$-LULUCF | Global (0.25 degree grid cells) | no split | 1700-2019 | Hansis et al. (2015), updated simulations described by Friedlingstein et al. (2020) | https://doi.org/10.18160/GCP-2020 |
| OSCAR – an Earth system compact model | OSCAR | 2020 | $CO_2$-LULUCF | Global (10 regions) | no split | 1701-2019 | Gasser et al. (2017, 2020); Friedlingstein et al. (2020) | https://doi.org/10.18160/GCP-2020 |
| Houghton and Nassikas Bookkeeping Model | H&N | 2020 | $CO_2$-LULUCF | Global (187 countries) | no split | 1850-2019 | Houghton and Nassikas (2017), Friedlingstein et al. (2020) | https://doi.org/10.18160/GCP-2020 |
| The Greenhouse gas – Air pollution INteractions and Synergies Model | GAINS | 2020 | $CO_2$, $CH_4$, $N_2O$, F-gases | Global (172 regions) | 3 main sectors, 16 subsectors | 1990-2015 | Höglund-Isaksson (2012, 2020), Winiwarter et al. (2018) | https://gains.iiasa.ac.at/models/index.html |
| EPA-Global Non-CO2 Greenhouse Gas Emissions | US-EPA | 2019 | $CH_4$, $N_2O$, F-gases: HFCs, PFCs, $SF_6$ | Global (195 countries) | 4 major sectors | 1990-2015 | EPA (2019) | https://www.epa.gov/global-mitigation-non-co2-greenhouse-gases |
| GCP – global nitrous oxide budget | GCP/INI | 2020 | $N_2O$ | 10 land regions and 3 oceanic regions | 21 natural and human sectors | 1980-2016 | Tian et al. (2020) | https://www.globalcarbonproject.org/nitrousoxidebudget/ |
| FAOSTAT – Emissions Totals | FAOSTAT | 2021 | $CO_2$, $CH_4$, $N_2O$ | Global (191 countries) | 15 activities in AFOLU | 1961-2019 | Frederici et al. (2015), Tubiello et al. (2013; 2021), Tubiello (2019) | http://www.fao.org/faostat/en/#data/GT |
| Fire Inventory from NCAR | FINN | | $CO_2$, $CH_4$, $N_2O$ | Global | | | Wiedinmyer et al. (2011) | |
| Global fire assimilation system | GFAS | | $CO_2$, $CH_4$, $N_2O$ | Global | | | Kaiser et al. (2012) | |

| Global fire emissions database | GFED | $CO_2$, $CH_4$, $N_2O$ | Global | | Van der Werf et al. (2017) | https://www.geo.vu.nl/~gwerf/GFED/GFED4/ |
| Quick fire emissions dataset | QFED | $CO_2$-LULUCF, $CH_4$, $N_2O$ | Global | | Darmenov and da Silva (2015) | |

## 2 Methods and Data

 ### 2.1 Overview

Our dataset provides a comprehensive, synthetic set of estimates for global GHG emissions disaggregated by 29 economic sectors and 228 countries. Our focus is on anthropogenic GHG emissions: natural sources and sinks are not included. We distinguish five groups of gases: (1) $CO_2$ emissions from fossil fuel combustion and industry ($CO_2$-FFI); (2) $CO_2$ emissions from land use, land-use change and forestry ($CO_2$-LULUCF); (3) methane emissions ($CH_4$); (4) nitrous oxide emissions ($N_2O$); (5) fluorinated gases (F-gases) comprising hydrofluorocarbons (HFCs), perfluorocarbons (PFCs), sulphur hexafluoride ($SF_6$) as well as nitrogen trifluoride ($NF_3$). F-gases that are internationally regulated as ozone depleting substances under the Montreal Protocol such as chlorofluorocarbons (CFCs) and hydrochlorofluorocarbons (HCFCs) are not included. We provide and analyse the GHG emissions data both in native units as well as in $CO_2$-equivalents ($CO_2$eq) (see Section 3.7) as commonly done in wide parts of the climate change mitigation community using global warming potentials with a 100 year time horizon from the IPCC Sixth Assessment Report (AR6) (Forster et al., 2021). We briefly discuss the impact of alternative metric choices in tracking aggregated GHG emissions over the past few decades and juxtapose this estimates of anthropogenic warming.

We report the annual growth rate in emissions $E$ for adjacent years (in percent per year) by calculating the difference between the two years and then normalizing to the emissions in the first year: $((E_{(t0+1)}-E_{t0})/E_{t0})\times100$. We apply a leap-year adjustment where relevant to ensure valid interpretations of annual growth rates. This affects the growth rate by about $0.3\%\,\text{yr}^{-1}$ (1/366) and causes calculated growth rates to go up by approximately 0.3% if the first year is a leap year and down by 0.3% if the second year is a leap year. We calculate the relative growth rate in percent per year for multi-year periods (e.g. a decade) by fitting a linear trend to the logarithmic transformation of E across time (see Friedlingstein et al., 2020).

We compile our dataset from four sources: (1) the full EDGARv6 release for $CO_2$-FFI as well as non-$CO_2$ GHGs covering the time period 1970-2018 (Crippa et al., 2021); (2) EDGARv6 fast-track data for $CO_2$-FFI providing preliminary estimates for 2019 (and 2020) (Crippa et al., 2021); (3) $CO_2$-LULUCF as the average of three bookkeeping models, consistent with the approach of the global carbon project (Friedlingstein et al., 2020). (4) 2019 non-$CO_2$ emissions based on Olivier and Peters (2020).

As shown in
Table 2, sectoral detail is organised along five major economic sectors as commonly used in IPCC reports on climate change mitigation (IPCC, 2014): energy supply, buildings, transport, industry as well as Agriculture, Forestry and Other Land-Use Changes (AFOLU). We devise a classification for assigning our 228 countries to regions, combining the standard Annex I/non-

Annex I distinction with geographical location. We provide other common regional classifications from the UN and the World Bank as part of the supplementary files. The dataset including the sector and region classification can be found at https://doi.org/10.5281/zenodo.5053055.

**Table 2 – Overview of the two-level sector aggregation with reference to assigned source/sink categories conforming to the IPCC reporting guidelines (IPCC, 2006, 2019) as well as relevant GHGs.** Note that EDGAR v6 distinguishes biogenic $CO_2$ and $CH_4$ sources with a "bio" label, with all other sectors "fossil" by default, even if that source is not related to fossil fuel activities. The fossil/bio label is hence not descriptive in nature. Two HCFC gases (denoted with *) are included in the dataset, despite being neither PFCs nor HFCs (and hence regulated under Montreal). This is to preserve consistency with current and previous versions of EDGAR, which include these gases.
Their total warming effect is low (~10 MtCO2eq in 2019) and the major HCFC sources are not included.

| Sector | Sub-sector | IPCC (2006) | Gases |
|---|---|---|---|
| **AFOLU (Agriculture, Forestry and Other Land-Use Changes)** | Biomass burning [agricultural waste burning on fields] | 3.C.1.b (bio) | $CH_4$, $N_2O$ |
| | Enteric Fermentation | 3.A.1.a.i (fossil), 3.A.1.a.ii (fossil), 3.A.1.b (fossil), 3.A.1.c (fossil), 3.A.1.d (fossil), 3.A.1.e (fossil), 3.A.1.f (fossil), 3.A.1.g (fossil), 3.A.1.h (fossil) | $CH_4$ |
| | Managed soils and pasture | 3.C.4 (fossil), 3.C.5 (fossil), 3.C.6 (fossil), 3.C.3 (fossil), 3.C.2 (fossil) | $CO_2$, $N_2O$ |
| | Manure management | 3.A.2.a.i (fossil), 3.A.2.a.ii (fossil), 3.A.2.b (fossil), 3.A.2.c (fossil), 3.A.2.i (fossil), 3.A.2.d (fossil), 3.A.2.e (fossil), 3.A.2.f (fossil), 3.A.2.g (fossil), 3.A.2.h (fossil) | $CH_4$, $N_2O$ |
| | Rice cultivation | 3.C.7 (fossil) | $CH_4$ |
| | Synthetic fertilizer application | 3.C.4 (fossil) | $N_2O$ |
| | Land-use change | | $CO_2$ |
| **Buildings** | Non-CO2 (all buildings) | 2.F.3 (fossil), 2.F.4 (fossil), 2.G.2.c (fossil) | c-C4F8, C4F10, CF4, HFC-125, HFC-227ea, HFC-23, HFC-236fa, HFC-134a, HFC-152a, SF6 |
| | Non-residential | 1.A.4.a (bio), 1.A.4.a (fossil) | $CO_2$, $CH_4$, $N_2O$ |
| | Residential | 1.A.4.b (bio), 1.A.4.b (fossil) | $CO_2$, $CH_4$, $N_2O$ |
| **Energy systems** | Coal mining fugitive emissions | 1.B.1.a (fossil), 1.B.1.c (fossil) | $CO_2$, $CH_4$ |
| | Electricity & heat | 1.A.1.a.i (bio), 1.A.1.a.i (fossil), 1.A.1.a.ii (bio), 1.A.1.a.ii (fossil), 1.A.1.a.iii (bio), 1.A.1.a.iii (fossil) | $CO_2$, $CH_4$, $N_2O$ |
| | Oil and gas fugitive emissions | 1.B.2.a.iii.2 (bio), 1.B.2.a.iii.2 (fossil), 1.B.2.a.iii.3 (fossil), 1.B.2.a.iii.4 (fossil), 1.B.2.b.iii.2 (fossil), 1.B.2.b.iii.4 (fossil), 1.B.2.b.iii.5 (fossil), 1.B.2.b.iii.3 (fossil), 1.B.2.b.ii (fossil), 1.B.2.a.ii (fossil) | $CO_2$, $CH_4$, $N_2O$ |
| | Other (energy systems) | 1.A.1.c.ii (bio), 1.A.1.c.ii (fossil), 1.A.1.c.i (bio), 1.A.1.c.i (fossil), 1.A.4.c.i (bio), 1.A.4.c.i (fossil), 1.A.5.a (bio), 1.A.5.a (fossil), 1.B.1.c (bio), 2.G.1.b (fossil), 5.B (fossil), 5.A (fossil) | $CO_2$, $CH_4$, $N_2O$, SF6 |

| | | | |
|---|---|---|---|
| | Petroleum refining | 1.A.1.b (bio), 1.A.1.b (fossil) | $CO_2$, $CH_4$, $N_2O$ |
| **Industry** | Cement | 2.A.1 (fossil) | $CO_2$ |
| | Chemicals | 1.A.2.c (bio), 1.A.2.c (fossil), 2.A.2 (fossil), 2.A.4.d (fossil), 2.A.4.b (fossil), 2.A.3 (fossil), 2.B.1 (fossil), 2.B.2 (fossil), 2.B.3 (fossil), 2.B.5 (fossil), 2.B.8.f (fossil), 2.B.8.b (fossil), 2.B.8.c (fossil), 2.B.8.a (fossil), 2.B.4 (fossil), 2.B.6 (fossil), 2.B.9.b (fossil), 2.D.3 (fossil), 2.G.3.a (fossil), 2.G.3.b (fossil) | $CO_2$, $CH_4$, $N_2O$, c-C4F8, C2F6, C3F8, C4F10, C5F12, C6F14, CF4, HFC-125, HFC-134a, HFC-143a, HFC-152a, HFC-227ea, HFC-32, HFC-365mfc, NF3, SF6, HFC-23 |
| | Metals | 1.A.1.c.i (fossil), 1.A.1.c.ii (fossil), 1.A.2.a (bio), 1.A.2.a (fossil), 1.A.2.b (bio), 1.A.2.b (fossil), 1.B.1.c (fossil), 2.C.1 (fossil), 2.C.2 (fossil), 2.C.3 (fossil), 2.C.4 (fossil), 2.C.5 (fossil), 2.C.6 (fossil) | $CO_2$, $CH_4$, $N_2O$, C2F6, CF4, SF6 |
| | Other industry | 1.A.2.d (bio), 1.A.2.d (fossil), 1.A.2.e (bio), 1.A.2.e (fossil), 1.A.2.f (bio), 1.A.2.f (fossil), 1.A.2.k (fossil), 1.A.2.i (fossil), 1.A.5.b.iii (fossil), 2.F.1.a (fossil), 2.F.2 (fossil), 2.F.5 (fossil), 2.E.1 (fossil), 2.E.2 (fossil), 2.E.3 (fossil), 2.G.1.a (fossil), 2.G.2.c (fossil), 2.G.2.b (fossil), 2.G.2.a (fossil), 2.D.1 (fossil), 5.A (fossil) | $CO_2$, $CH_4$, $N_2O$, HFC-125, HFC-134a, HFC-143a, HFC-152a, HFC-227ea, HFC-236fa, HFC-245fa, HFC-32, HFC-365mfc, C3F8, C6F14, CF4, HFC-43-10-mee, HFC-134, HFC-143, HFC-23, HFC-41, c-C4F8, C2F6, NF3, SF6, HCFC-141b*, HCFC-142b*, C4F10 |
| | Waste | 4.A.1 (fossil), 4.D.2 (fossil), 4.D.1 (fossil), 4.C.1 (fossil), 4.C.2 (bio), 4.C.2 (fossil), 4.B (fossil) | $CO_2$, $CH_4$, $N_2O$ |
| **Transport** | Domestic Aviation | 1.A.3.a.ii (fossil) | $CO_2$, $CH_4$, $N_2O$ |
| | Inland Shipping | 1.A.3.d.ii (bio), 1.A.3.d.ii (fossil) | $CO_2$, $CH_4$, $N_2O$ |
| | International Aviation | 1.A.3.a.i (fossil) | $CO_2$, $CH_4$, $N_2O$ |
| | International Shipping | 1.A.3.d.i (bio), 1.A.3.d.i (fossil) | $CO_2$, $CH_4$, $N_2O$ |
| | Other (transport) | 1.A.3.e.i (bio), 1.A.3.e.i (fossil), 1.A.4.c.ii (fossil), 1.A.4.c.iii (bio), 1.A.4.c.iii (fossil) | $CO_2$, $CH_4$, $N_2O$ |
| | Rail | 1.A.3.c (bio), 1.A.3.c (fossil) | $CO_2$, $CH_4$, $N_2O$ |
| | Road | 1.A.3.b (bio), 1.A.3.b (fossil) | $CO_2$, $CH_4$, $N_2O$ |

## 2.2 The Emissions Database for Global Atmospheric Research (EDGAR)

EDGAR emission estimates included in our dataset are derived from the full version 6 release which includes $CO_2$ and non-$CO_2$ GHG emission estimates from 1970 to 2018 computed from stable international statistics and on fast-track estimates of fossil CO2 emissions up to the year 2020 (Crippa et al., 2021). This general EDGAR methodological description is largely taken from Janssens-Maenhout et al. (2019). The EDGAR bottom-up emission inventory estimates are calculated from international activity data and emission factors following the 2006 IPCC Guidelines for National Greenhouse Gas Inventories (IPCC, 2006) - updated according to the latest scientific knowledge. Emissions (EMs) from a given sector $i$ in a country C accumulated during a year $t$ for a chemical compound x are calculated with the country-specific activity data (AD), quantifying the activity in sector $i$, with the mix of $j$ technologies (TECH) and with the mix of $k$ (end-of-pipe) abatement measures (EOP)

installed with the share $k$ for each technology $j$, the emission rate with an uncontrolled emission factor (EF) for each sector $i$ and technology $j$ and relative reduction (RED) by abatement measure $k$, as summarised in the following formula:

$$EM_i(C, t, x) = \sum_{j,k} \left[ AD_i(C, t) \cdot TECH_{i,j}(C, t) \cdot EOP_{i,j,k}(C, t) \cdot EF_{i,j}(C, t, x) \cdot \left(1 - RED_{i,j,k}(C, t, x)\right) \right]$$

The activity data are sector dependent and vary from fuel combustion in energy units (TJ) of a particular fuel type, to the amount (ton) of products manufactured, or to the number of animals or the area (hectares) or yield (ton) of cultivated crops. The technology mixes, (uncontrolled) emission factors and end-of-pipe measures are determined at different levels: country-specific, regional, country group (e.g. Annex I/non-Annex I), or global. Technology-specific emission factors are used to enable an IPCC tier-2 approach, taking into account the different management and /technology processes or infrastructures

(e.g., different distribution networks) under specific "technologies", and modelling explicitly abatements/ emission reductions, e.g. the $CH_4$ recovery from coal mine gas at country level under the "end-of-pipe measures". As with national inventories, emissions are accounted over a period of one calendar year in the country in which they took place (i.e. a territorial accounting principle) (IPCC, 2006, 2019). A full description of data sources and methodology for EDGARv6 is provided in Crippa et al. (2021).


To compute emissions up to most recent years, a Fast-Track methodology is applied, as described in detailed in Oreggioni et al. (2021). The underlying idea is to extrapolate trends based on observed activity patterns in representative sectors. For $CO_2$-FFI emissions, the fast track estimates were based on the latest BP coal, oil and natural gas consumption data (BP, 2021). Emission updates for cement, lime, ammonia and ferroalloys production beyond 2018 are still based on stable statistics and in

particular on US Geological Survey statistics, urea production and consumption on statistics from the International Fertilizer Association, gas used from flaring on data from the Global Gas Flaring Reduction Partnership, steel production on statistics from the World Steel Association, and cement clinker production on UNFCCC data. Fast-track extensions for non-$CO_2$ GHG emissions are developed from Olivier and Peters (2020). For $CH_4$ and $N_2O$ these are based on agricultural statistics from Food and Agricultural Organization (FAO) ($CH_4$ and $N_2O$) of the United Nations, fuel production and transmission statistics from

IEA and BP ($CH_4$) as well as UNFCCC-CRF data for Annex I countries on coal production ($CH_4$ recovery) and the production of chemicals ($N_2O$ abatement). Finally, for F-gases the fast-track extension was based on the most recent national emission inventories, submitted under the UNFCCC (up to 2018). For all remaining countries and years, a simple extrapolation was used given the absence of international statistics. We apply this fast-track data by Olivier and Peters (2020) to our dataset by calculating the county and sector specific emissions growth between 2018 and 2019 and multiplying it with the 2018 values

in our data.

## 2.3 Accounting for $CO_2$ emissions Land Use, Land-Use Change and Forestry ($CO_2$-LULUCF)

We consider all fluxes of $CO_2$ from land use, land-use change and forestry. This includes $CO_2$ fluxes from the clearing of forests and other natural vegetation (by anthropogenic fire and/or clear-cut), afforestation, logging and forest degradation (including harvest activity), shifting cultivation (cycles of forest clearing for agriculture, then abandonment), and regrowth of forests and other natural vegetation following wood harvest or abandonment of agriculture, and emissions from peat burning and drainage. Some of these activities lead to emissions of $CO_2$ to the atmosphere, while others lead to $CO_2$ sinks. $CO_2$-LULUCF therefore is the net sum of emissions and removals from all human-induced land use changes and land management. Note that $CO_2$-LULUCF is referred to as (net) land-use change emissions, ELUC, in the context of the global carbon budget (Friedlingstein et al., 2020). Agriculture per se, apart from conversions between different agricultural types, does not lead to substantial $CO_2$ emissions as compared to land-use changes such as clearing or regrowth of natural vegetation. Therefore, $CO_2$ fluxes in the AFOLU sector refer mostly to forestry and other land use (changes), while the agricultural part of the sector is mainly characterized by $CH_4$ and $N_2O$ fluxes.

Since in reality anthropogenic $CO_2$-LULUCF emissions co-occur with natural $CO_2$ fluxes in the terrestrial biosphere, models have to be used to distinguish anthropogenic and natural fluxes (Friedlingstein et al., 2020). $CO_2$-LULUCF as reported here is calculated via a bookkeeping approach, as originally proposed by Houghton et al. (2003), tracking carbon stored in vegetation and soils before and after land-use change. Response curves are derived from the literature and observations to describe the temporal evolution of the decay and regrowth of vegetation and soil carbon pools for different ecosystems and land use transitions, including product pools of different lifetimes. These dynamics distinguish bookkeeping models from the common approach of estimating "committed emissions" (assigning all present and future emissions to the time of the land use change event), which is frequently derived from remotely-sensed land use area or biomass observations (Ramankutty et al., 2007). Most bookkeeping models also represent the long-term degradation of primary forest as lowered standing vegetation and soil carbon stocks in secondary forests, and include forest management practices such as wood harvesting.

The scientific definition of $CO_2$-LULUCF emissions used here differs from the one applied in national greenhouse gas inventories (GHGI) or the inventory data provided by FAOSTAT (Tubiello et al., 2021). Concretely, this means that inventory data include natural terrestrial fluxes caused by changes in environmental conditions, e.g., effects of rising atmospheric $CO_2$ ("$CO_2$-fertilization"), climate change, and nitrogen deposition -- sometimes called "indirect effects" as opposed to the direct anthropogenic effects of land-use change and management (Houghton et al., 2012) - when they occur on area that countries declare as managed. Since environmental changes turned the terrestrial biosphere into a massive sink, removing about one third of annual anthropogenic emissions in the last decade (Friedlingstein et al., 2020), it is unsurprising that global emission estimates are smaller for inventory data than for the scientific definition (see Figure 1). About 3.2 $GtCO_2$ $yr^{-1}$ (for the period

2005-2014) was found to be explicable by these conceptual differences in anthropogenic forest sink estimation related to the representation of environmental change impacts and the areas considered as managed (Grassi et al., 2018).

These two conceptually different approaches have different aims: The scientific approach separates natural from anthropogenic drivers, i.e., effects of changes in environmental conditions from effects of land-use change and land management. By contrast, the inventory approach separates fluxes based on areas, with all those occurring on managed land being declared anthropogenic. Given that observational data of carbon stocks or fluxes cannot distinguish the co-occurring effects of environmental changes and land-use activities, an area-based approach that does not require this distinction can more consistently be implemented across countries. These conceptual differences between scientific and inventory approaches have been acknowledged (Canadell et al., 2021; Petrescu et al., 2020a) and approaches have been developed to map the scientific and inventory definitions to each other (Grassi et al., 2018, 2021). For non-$CO_2$ GHGs, drivers and areas coincide, such that FAOSTAT data for $CH_4$ and $N_2O$ is complementary to bookkeeping $CO_2$-LULUCF emissions.

Following the approach taken by the global carbon budget (Friedlingstein et al., 2020), we take the average of three bookkeeping estimates: the bookkeeping of land use emissions model, BLUE (Hansis et al., 2015), H&N (Houghton and Nassikas, 2017), and OSCAR (Gasser et al., 2020). Key differences across these estimates, including land-use forcing, are summarised in Table 4. Since bookkeeping models do not include emissions from organic soils, emissions from peat fires and peat drainage are added from external datasets: Peat burning is based on the Global Fire Emission Database (GFED4s; van der Werf et al., 2017) and introduces large interannual variability to the $CO_2$-LULUCF emissions due to synergies of land-use and climate variability particularly in Southeast Asia, strongly noticeable during El-Niño events such as 1997. Peat drainage is based on estimates by Hooijer et al. (2010) for Indonesia and Malaysia in H&N, and added to BLUE and OSCAR from the global FAO data on organic soils emissions from croplands and grasslands (Conchedda and Tubiello, 2020).

## 3. Uncertainties in GHG emission estimates

Estimates of historic GHG emissions – $CO_2$, $CH_4$, $N_2O$ and F-gases – are uncertain to different degrees. Assessing and reporting uncertainties is crucial in order to understand whether available estimates are sufficiently accurate to answer, for example, whether GHG emissions are still rising, or if a country has achieved an emission reduction goal (Marland, 2008). These uncertainties can be of a scientific nature, such as when a process is not sufficiently understood. They also arise from incomplete or unknown parameter information (activity data, emission factors etc.), as well as estimation uncertainties from imperfect modelling techniques. There are at least three major ways to examine uncertainties in emission estimates (Marland et al., 2009): 1) by comparing estimates made by independent methods and observations (e.g. comparing top-down vs bottom-up estimates; modelling against remote sensing data) (Petrescu et al., 2020a, 2021a, 2021b; Saunois et al., 2020a; Tian et al., 2020) (Petrescu et al., 2020b, 2020a; Saunois et al., 2020b; Tian et al., 2020); 2) by comparing estimates from multiple sources

and understanding sources of variation (Andres et al., 2012; Andrew, 2020a; Ciais et al., 2021; Macknick, 2011); 3) by evaluating multiple estimates from a single source (e.g. Hoesly and Smith, 2018) including approaches such as uncertainty ranges estimated through statistical sampling across parameter values, applied for example at the country or sectoral level (e.g. Andres et al., 2014; Monni et al., 2007; Solazzo et al., 2021), or to spatially distributed emissions (Tian et al., 2019).

Uncertainty estimates can be rather different depending on the method chosen. For example, the range of estimates from multiple sources is bounded by their interdependency; they can be lower than true structural plus parameter uncertainty estimates or than estimates made by independent methods. In particular it is important to account for potential bias in estimates, which can result from using common methodological or parameter assumptions across estimates, or from missing sources, which can result in a systemic bias in emission estimates (see $N_2O$ discussion below). Independent top-down observational constraints are, therefore, particularly useful to bound total emission estimates (Petrescu et al., 2021b, 2021a).

Solazzo et al. (2021) evaluated the uncertainty of the EDGAR's source categories and totals for the main GHGs ($CO_2$-FFI, $CH_4$, $N_2O$). This study is based on the propagation of the uncertainty associated with input parameters (activity data and emission factors) as estimated by expert judgement (tier-1) and complied by IPCC (2006, 2019). A key methodological challenge is determining how well uncertain parameters are correlated between sectors, countries, and regions. The more highly correlated parameters (e.g. emission factors) are across scales, the higher the resulting overall uncertainty estimate. Solazzo et al. (2021) assume full covariance between same source categories where similar assumptions are being used, and independence otherwise. For example, they assume full covariance where the same emission factor is used between countries or sectors, while assuming independence where country-specific emission factors are used. This strikes a balance between extreme assumptions (full independence or full covariance in all cases) that are likely unrealistic, but still leans towards higher uncertainty estimates. When aggregating emission sources, assuming covariance increases the resulting uncertainty estimate. Uncertainties calculated with this methodology tend to be higher than the range of values from ensemble of dependent inventories (Saunois et al., 2016, 2020b). The uncertainty of emission estimates derived from ensembles of gridded results from bio-physical models (Tian et al., 2018) adds an additional dimension of spatial variability, and is therefore not directly comparable with aggregate country or regional uncertainty, estimated with the methods discussed above.

This section provides an assessment of uncertainties in greenhouse gas emissions data at the global level. The uncertainties reported here combine statistical analysis, comparisons of global emissions inventories and expert judgement of the likelihood of results lying outside a defined confidence interval, rooted in an understanding gained from the relevant literature. At times, we also use a qualitative assessment of confidence levels to characterize the annual estimates from each term based on the type, amount, quality, and consistency of the evidence as defined by the IPCC (2014).

Such a comprehensive uncertainty assessment covering all major groups of greenhouse gases and considering multiple lines of evidence has been missing in the literature. The absence has provided a serious challenge for a transparent, scientific reporting of GHG emissions in climate change assessments like those by IPCC's Working Group III or the UN Emissions Gap Report that have only more recently started to even deal with the issue (Blanco et al., 2014; UNEP, 2020). Most of the available studies in the peer-reviewed literature using multiple lines of evidence for their assessment have focused on individual gases like in the Global Carbon Budget (Friedlingstein et al., 2020), the Global Methane Budget (Saunois et al., 2020b) or the Global Nitrous Oxide Budget (Tian et al., 2020) or covered multiple gases, but mainly considered individual lines of evidence (Janssens-Maenhout et al., 2019; Solazzo et al., 2021).

We adopt a 90% confidence interval (5[th]-95[th] percentile) to report the uncertainties in our GHG emissions estimates, i.e., there is a 90 % likelihood that the true value will be within the provided range if the errors have a Gaussian distribution, and no bias is assumed. This is in line with previous reporting in IPCC AR5 (Blanco et al., 2014; Ciais et al., 2014). We note that national emissions inventory submissions reported to the UNFCCC are requested to report uncertainty using a 95% or $2\sigma$ confidence interval. The use of this broader uncertainty interval implies, however, a relatively high degree of knowledge about the uncertainty structure of the associated data, particularly regarding the distribution of uncertainty in the tails of the probability distributions. Such a high degree of knowledge is not present over all the emission sectors and species considered here. Note that in some cases below we convert $1\sigma$ uncertainty results from the literature to a 90% confidence interval by implicitly assuming a normal distribution. While we do this as a necessary assumption to obtain a consistent estimate across all GHGs, we note that this itself is an assumption that may not be valid. We have made use of the best available information in the literature, but note that much more work on uncertainty quantification remains to be done. Using IPCC uncertainty language, we cannot assign *high confidence* to the robustness of most existing uncertainty estimates.

### 3.1 CO₂ emissions from fossil fuels and industrial processes

Several studies have compared estimates of annual $CO_2$-FFI emissions from different global inventories (Andres et al., 2012; Andrew, 2020a; Gütschow et al., 2016; Janssens-Maenhout et al., 2019; Macknick, 2011; Petrescu et al., 2020b). However, estimates are not fully independent as they all ultimately rely on many of the same data sources. For example, all global inventories use one of four global energy datasets to estimate $CO_2$ emissions from energy use, and these energy datasets themselves all rely on the same national energy statistics, with few exceptions (Andrew, 2020a). Some divergence between these estimates (see Figure 1) are related to differences in the estimation methodology, conversion factors, emission coefficients, assumptions about combustion efficiency, and calculation errors (Andrew, 2020a; Marland et al., 2009). Key differences for nine global datasets are highlighted in Table 3 (see also Table 1 for further information on the inventories). Another important source of divergence between datasets is differences in their respective system boundaries (Andres et al., 2012; Andrew, 2020a; Macknick, 2011). Hence, differences across $CO_2$-FFI emissions estimates do not reflect full uncertainty

due to source data dependencies. At the same time, the observed range across estimates from different databases exaggerates uncertainty, to the extent that they largely originate in system boundary differences (Andrew, 2020a; Macknick, 2011).

**Table 3 - System boundaries and other key features of global FFI-CO₂ emissions datasets as published.** Comparison of some important general characteristics of nine emissions datasets, with green indicating a characteristic that might be considered a strength. Columns four to six refer to $CO_2$ emission estimates for industrial processes and product use. Since all datasets are under development, these details are subject to change. Further information on the individual inventories can be found in Table 1. Based on Andrew (Andrew, 2020a)

| | Primary source | Uses IPCC emission factors | Includes venting & flaring | Includes cement | Includes other carbonates | Non-fuel use based on | Reports bunkers separately | By fuel type | By sector | Includes official estimates |
|---|---|---|---|---|---|---|---|---|---|---|
| CDIAC | yes | no | yes | yes | no | national data | yes | yes | no | no |
| BP | yes | yes | no | no | no | national data | no | no | no | no |
| IEA | yes | yes | no | no | no | national data | yes | yes | yes | no |
| EDGAR | yes | yes | yes | yes | yes | national data | yes | no | yes | no |
| EIA | yes | no | yes | no | no | US data | no | yes | no | no |
| GCP | partial | no | yes | yes | partial | national data | yes | yes | no | yes |
| CEDS | mostly | no | yes | yes | yes | national data | yes | yes | yes | yes |
| PRIMAP-hist | no | no | yes | yes | yes | national data | yes | no | yes | yes |
| UNFCCC CRFs | yes | partial | yes | yes | yes | national data | yes | yes | yes | yes |

Across global inventories, mean global annual $CO_2$-FFI emissions track at 34.4±2 $GtCO_2$ in 2014, reflecting a variability of about ±5.4% (Figure 1). However, this variability is almost halved when system boundaries are harmonised (Andrew, 2020a). EDGARv6 $CO_2$-FFI emissions as used in this report track at the top of the range as shown in Figure 1. This is partly due to the comprehensive system boundaries of EDGAR, but also due to the assumption of 100% oxidation of combusted fuels as per IPCC default assumptions. Once system boundaries are harmonised EDGAR continues to track at the upper end of the range, but no longer at the top. EDGAR $CO_2$-FFI estimates are further well-aligned with emission inventories submitted by Annex I countries to the UNFCCC – even though some variation can occur for individual countries such as Kazakhstan, Ukraine or Estonia, in general, or for certain years (see Figure SM-4). Differences in FFI-$CO_2$ emissions across different versions of the EDGAR dataset are shown in the Supplementary Material (see Figure SM-1).

Uncertainties in $CO_2$-FFI emissions arise from the combination of uncertainty in activity data and uncertainties in emission factors including assumptions for combustion completeness and non-combustion uses. $CO_2$-FFI emissions estimates are largely derived from energy consumption activity data, where data uncertainties are comparatively small due to well established statistical monitoring systems, although there are larger uncertainties in some countries and time periods (Andres et al., 2012; Andrew, 2020a; Ballantyne et al., 2015; Janssens-Maenhout et al., 2019; Macknick, 2011). Most of the underlying uncertainties are systematic and related to underlying biases in the energy statistics and accounting methods used (Friedlingstein et al., 2020). Uncertainties are lower for fuels with relatively uniform properties such as natural gas, oil or gasoline and higher for fuels with more diverse properties, such as coal (IPCC 2006; Blanco G. et al. 2014). Uncertainties in $CO_2$ emissions estimates from industrial processes, i.e. non-combustive oxidation of fossil fuels and decomposition of carbonates, are higher than for fossil fuel combustion. At the same time, products such as cement also take up carbon over their life cycle, which are often not fully considered in carbon balances (Guo et al., 2021; Sanjuán et al., 2020; Xi et al., 2016). However, recent versions of the global carbon budget include specific estimates for the cement carbonation sink and estimate average annual $CO_2$ uptake at 0.70 $GtCO_2$ for 2010-2019 (Friedlingstein et al., 2020).

Uncertainties of energy consumption data (and, therefore, $CO_2$-FFI emissions) are generally higher for the first year of their publication when less data is available to constrain estimates. In the BP energy statistics, 70% of data points are adjusted by an average of 1.3% of a country's total fossil fuel use in the subsequent year with further more modest revisions later on (Hoesly and Smith, 2018). Uncertainties are also higher for developing countries, where statistical reporting systems do not have the same level of maturity as in many industrialised countries (Andres et al., 2012; Andrew, 2020b; Friedlingstein et al., 2019, 2020; Gregg et al., 2008; Guan et al., 2012; Janssens-Maenhout et al., 2019; Korsbakken et al., 2016; Marland, 2008). Example estimates of uncertainties for $CO_2$ emissions from fossil fuel combustion at the 95% confidence interval are ±3-5% for the U.S., ±15 - ±20% for China and ±50% or more for countries with poorly developed or maintained statistical infrastructure (Andres et al., 2012; Gregg et al., 2008; Marland et al., 1999). However, these customary country groupings do not always predict the extent to which a country's energy data has undergone historical revisions (Hoesly and Smith, 2018). Uncertainties in $CO_2$-FFI emissions before the 1970s are higher than for more recent estimates. Over the last two to three decades uncertainties have increased again because of increased production in some developing countries with less rigorous statistics and more uncertain fuel properties (Ballantyne et al., 2015; Friedlingstein et al., 2020; Marland et al., 2009).

The global carbon project (Friedlingstein et al., 2019, 2020; Le Quéré et al., 2018) assesses uncertainties in global anthropogenic $CO_2$-FFI emissions estimates within one standard deviation ($1\sigma$) as ±5% (±10% at $2\sigma$). This is broadly consistent with the ±8.4% uncertainty estimate for CDIAC (Andres et al., 2014) as well as the ±7 - ±9% uncertainty estimate for EDGARv4.3.2 and v5 (Janssens-Maenhout et al., 2019; Solazzo et al., 2021) at $2\sigma$. It remains at the higher end of the ±5% - ±10% range provided by Ballantyne et al. (2015). Consistent with the above uncertainty assessments, we present uncertainties

for global anthropogenic CO₂ emissions at ±8% for a 90% confidence interval in line with IPCC AR5 and the UN emissions gap report (Blanco G. et al., 2014; UNEP, 2020).

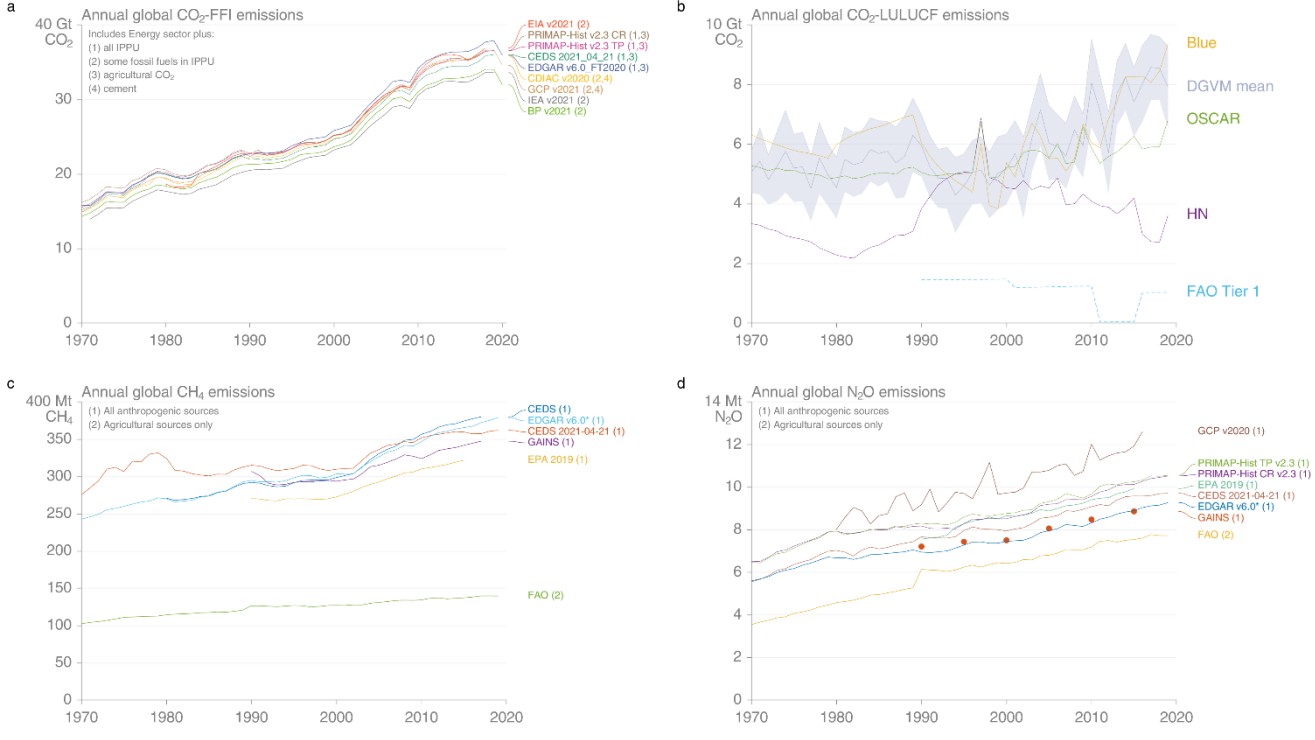

**Figure 1 - Estimates of global anthropogenic GHG emissions from different data sources 1970-2019.** Top-left panel: CO₂ FFI emissions from: EDGAR - Emissions Database for Global Atmospheric Research (this dataset) (Crippa et al., 2021); GCP – Global Carbon Project (Andrew and Peters, 2021; Friedlingstein et al., 2020); CEDS - Community Emissions Data System (Hoesly et al., 2018; O'Rourke et al., 2021); CDIAC Global, Regional, and National Fossil-Fuel CO₂ Emissions (Gilfillan et al., 2020); PRIMAP-hist - Potsdam Real-time
Integrated Model for probabilistic Assessment of emissions Paths (Gütschow et al., 2016, 2021b); EIA - Energy Information Administration International Energy Statistics (EIA, 2021); BP - BP Statistical Review of World Energy (BP, 2021); IEA - International Energy Agency (IEA, 2021a, 2021b); IPPU refers to emissions from industrial processes and product use. Top-right panel: CO₂-LULUCF emissions from: BLUE – Bookkeeping of land-use emissions (Friedlingstein et al., 2020; Hansis et al., 2015); DGVM-mean – Multi-model mean of CO₂-LULUCF emissions from dynamic global vegetation models (Friedlingstein et al., 2020); OSCAR – an earth system compact model
(Friedlingstein et al., 2020; Gasser et al., 2020); HN – Houghton and Nassikas Bookkeeping Model (Friedlingstein et al., 2020; Houghton and Nassikas, 2017); for comparison, the net CO₂ flux from FAOSTAT is plotted, which comprises emissions and removals by forest land (FAOSTAT, 2021; Tubiello et al., 2021) (in contrast to the scientific definition this includes the natural terrestrial sink if occurring on managed land) and emissions from drained histosols under cropland/grassland (Conchedda and Tubiello, 2020). Bottom-left panel: Anthropogenic CH₄ emissions from: EDGAR (above); CEDS (above); GAINS - The Greenhouse gas – Air pollution Interactions and
Synergies Model (Höglund-Isaksson et al., 2020); EPA-2019: Greenhouse gas emission inventory (US-EPA, 2019); FAO –FAOSTAT inventory emissions (FAOSTAT, 2021; Tubiello, 2018; Tubiello et al., 2013); Bottom-right panel: Anthropogenic N₂O emissions from: GCP – global nitrous oxide budget (Tian et al., 2020); CEDS (above); EDGAR (above); GAINS (Winiwarter et al., 2018); EPA-2019 (above); FAO (above). Differences in emissions across different versions of the EDGAR dataset are shown in the Supplementary Material (Fig. SM-1).

## 3.2 Anthropogenic CO$_2$ emissions from land use, land use change and forestry (CO$_2$-LULUCF)

CO$_2$-LULUCF emissions are drawn from three global bookkeeping models. For 1990-2019, average net CO$_2$-LULUCF emissions are estimated at 6.1, 4.3, and 5.6 GtCO$_2$ yr$^{-1}$ for BLUE, H&N, and OSCAR (Friedlingstein et al., 2020). Gross emissions 1990-2019 for BLUE, H&N, OSCAR are 17, 9.6 and 19 GtCO$_2$ yr$^{-1}$, while gross removals are 11, 5.3, 13 GtCO$_2$ yr$^{-1}$ respectively. For 1990-2019 maximum average differences are 9.1 and 7.8 GtCO$_2$ yr$^{-1}$ for gross emissions and removals, respectively (Friedlingstein et al., 2020). Note that 2016-2019 is extrapolated in H&N and 2019 in OSCAR based on the anomalies of the net flux for the gross fluxes. Differences in the models underlying this observed variability are reported in Table 4. In the longer term, a consistent general upward trend since 1850 across models is reversed during the second part of the 20th century. Since the 1980s, however, differing trends across models are related to, among other things, different land-use forcings (Gasser et al., 2020). Further differences between BLUE and H&N can be traced in particular to: (1) differences in carbon densities between natural and managed vegetation, or between primary and secondary vegetation; (2) a higher allocation of cleared and harvested material to fast turnover pools in BLUE compared to H&N; and (3) to the inclusion sub-grid scale transitions (Bastos et al., 2021).

Uncertainties in CO$_2$-LULUCF emissions can be more comprehensively assessed through comparisons across a suite of dynamic global vegetation models (DGVM) (Friedlingstein et al., 2020). DGVM models are not combined in the CO$_2$-LULUCF mean estimate in our data because the typical DGVM setup includes the loss of additional sink capacity, i.e. the additional sink capacity forests could have provided in response to environmental changes, in particular the rise in CO$_2$, due to their long-lived biomass, but that is lost because large areas of forest were historically cleared for agriculture. The loss of additional sink capacity makes up about 40% of the DGVM estimate in recent years (Obermeier et al., 2020) and is excluded in bookkeeping estimates. Nonetheless, a CO$_2$-LULUCF estimate from the DGVM multi-model mean remains consistent with the average estimate from the bookkeeping models, as shown in Figure 1. Variation across DGVMs is large with a standard deviation at around 1.8 GtCO$_2$ yr$^{-1}$, but is still smaller than the average difference between bookkeeping models at 2.6 GtCO$_2$ yr$^{-1}$ as well as the current estimate of H&N (Houghton and Nassikas, 2017) and its previous model versions (Houghton et al., 2012). DGVMs differ in methodology, input data and how comprehensively they represent land-use-related processes. In particular land management, such as crop harvesting, tillage, or grazing (all implicitly included in observation-based carbon densities of bookkeeping models) can alter CO$_2$ flux estimates substantially, but are included to varying extents in DGVMs, thus increasing model spread (Arneth et al., 2017). For all types of models, land-use forcing is a major determinant of emissions and removals, and its high uncertainty impacts CO$_2$-LULUCF estimates (Bastos et al., 2021). The reconstruction of land-use change of the historical past, which has to cover decades to centuries of legacy LULUCF fluxes, is based on sparse data or proxies (Hurtt et al., 2020; Klein Goldewijk et al., 2017), while satellite-based products suffer from complications in distinguishing natural from anthropogenic drivers (Hansen et al., 2013; Li et al., 2018) or accounting for small-scale

disturbances and degradation (Matricardi et al., 2020). Lastly, regional carbon budgets can be substantially over- or underestimated when the carbon embodied in trade products is not accounted for (Ciais et al., 2021).

We base our uncertainty assessment on Friedlingstein et al. (2020) and take $\pm2.6$ GtCO$_2$ yr$^{-1}$ as a best-value judgement for the $\pm1\sigma$ uncertainty range (thus $\pm5.1$ GtCO$_2$ yr$^{-1}$ for $\pm2\sigma$) in CO$_2$-LULUCF emissions, constant over the last decades. This absolute uncertainty estimate presented above corresponds roughly to a relative uncertainty of about $\pm50\%$ over 1970-2019, which is much higher than for most fossil-emission terms, but reflects the large model spread and large differences between the current estimate of H&N and its previous model version (Houghton et al., 2012). This corresponds to a relative uncertainty of about

$\pm80\%$ for a 90% confidence interval (5$^{th}$-95$^{th}$ percentile) and is larger but still broadly in line with the upper end of the relative uncertainty of $\pm50 - \pm75\%$ considered in AR5 (Blanco G. et al., 2014). Much larger uncertainties in CO$_2$-LULUCF emissions have been identified across the literature, but were traced back to different definitions used in various modelling frameworks (Pongratz et al., 2014) as well as inventory data (Grassi et al., 2018). The constant absolute uncertainty estimate used in the global carbon budget (Friedlingstein et al., 2020) translates to a slightly lower relative uncertainty estimate, given that the

mean of the CO$_2$-LULUCF estimates has been increasing over the last few decades. Here we opt for a relative uncertainty estimate of $\pm70\%$ for a 90% confidence interval, which provides absolute uncertainty estimates for recent years that are consistent in magnitude with Friedlingstein et al. (2020).

Uncertainties can be much higher at a national level than at global level, since regional biases tend to cancel out. Land-use

forcing has been identified as major driver of differences at regional and global level (Gasser et al., 2020; Hartung et al., 2021; Rosan et al., 2021), as have assumptions on carbon densities and the allocation of cleared or harvested material to slash or product pools of various lifetimes, for which accurate global data over long time periods is missing (Bastos et al., 2021). Although the bookkeeping models are conceptually similar, the bookkeeping estimates include country-specific information to different extents: for example, fire suppression (for the U.S.) is included in H&N (Houghton and Nassikas, 2017), but not

the other estimates, and H&N includes peat drainage emissions only for Southeast Asia, while the FAO emissions estimates for organic soil drainage added to BLUE and OSCAR cover all countries (Friedlingstein et al., 2020). The effect of smoothing the FAO cropland and pasture information, which can be very variable in some countries, with a 5-year running mean in H&N, while the annual data is used for the recent decades in HYDE underlying BLUE and OSCAR, must also be expected to contribute to the spread in estimates on a country level. Overall, great care has to be taken when comparing estimates of

individual countries across models to not over-interpret differences.

Finally, note that attempts to constrain the estimates of CO$_2$-LULUCF emissions by observed biomass densities have been undertaken, but were successful only in some non-tropical regions (Li et al., 2017). While providing valuable independent and observation-driven information, remote-sensing derived estimates have limited applicability for model evaluation for the total

CO$_2$-LULUCF flux, since they usually only quantify vegetation biomass changes and exclude legacy emissions from the pre-

satellite era. Further, with the exception of the (pan-tropical) estimates by Baccini et al. (2012) they either track committed instead of actual emissions (e.g. Tyukavina et al., 2015), combine a static carbon density map with forest cover changes, or include the natural land sink (e.g. Baccini et al., 2017) to infer fluxes directly from the carbon stock time series – none of which fully distinguishes natural from anthropogenic disturbances.


**Table 4 - Key differences between global bookkeeping estimates for $CO_2$-LULUCF emissions.** Notes: DGVM – dynamic global vegetation model; LUH2 and FAO refer to land-use forcing datasets; arrows indicate tendency of process to increase or decrease emissions compared to the other estimates' choice.

| | Bookkeeping model | | |
|---|---|---|---|
| | **BLUE[a]** | **H&N[b]** | **OSCAR[c]** |
| **Geographical scale of computation** | 0.25 degree gridscale | country | 10 regions and 5 biomes |
| **Carbon densities of soil and vegetation** | literature-based | based on country reporting | calibrated to DGVMs |
| **Land-use forcing** | LUH2[d,e] | FAO[f] | LUH2 and FAO[d,e,f] |
| **Representation of processes (Arrows: indicative effect on $CO_2$-LULUCF emissions)** | | | |
| *Sub-grid scale ("gross") land-use transitions* | yes (↑) | no (↓) | yes (↑) |
| *Pasture conversion* | From all natural vegetation types proportionally (↑) | from grasslands first (↓) | from all natural vegetation types proportionally (↑) |
| *Distinction rangeland vs pasture[g]* | yes (↓) | no (↑) | no (↑) |
| *Coverage peat drainage (as in Global Carbon Budget 2020)* | World (↑)[h] | South East Asia (↓)[i] | World (↑)[h] |

Literature: [a] (Hansis et al., 2015); [b] (Houghton and Nassikas, 2017); [c] (Gasser et al., 2020); [d] (Hurtt et al., 2020); [e] (Chini et al., 2020); [f] (Nations, 2015);[g] based on rangeland-pasture distinction of the HYDE dataset (Klein Goldewijk et al., 2017) and forest cover map of Hurtt et al. (2020); see Friedlingstein et al. (2020) for details [h] (Conchedda and Tubiello, 2020); [i] (Hooijer et al., 2010)


### 3.3 Anthropogenic $CH_4$ emissions

About 60% of total global $CH_4$ emissions come from anthropogenic sources (Saunois et al., 2020b). These are linked to a range of different sectors: agriculture, fossil production and use, waste as well as biomass and biofuel burning. Methane emissions can be derived either using bottom-up (BU) estimates that rely on anthropogenic inventories such as EDGAR (Janssens-

Maenhout et al., 2019), land surface models that infer part of natural emissions (Wania et al., 2013) or observation-based upscaling for some specific sources such as geological sources (e.g. Etiope et al., 2019). Alternatively, top-down (TD) approaches can be used, such as atmospheric transport models that assimilate methane atmospheric observations to estimate past methane emissions (Houweling et al., 2017). Some TD systems aim to optimize certain emission sectors based on differences in their spatial and temporal distributions (e.g. Bergamaschi et al., 2013), while other only solve for net emissions

at the surface. Then the partitioning of TD posterior (output) fluxes between specific source sectors (e.g. *Fossil* vs. *BB&F*) is carried out with various degrees of uncertainty depending of the methods and the degree of refinement of sectors, but often rely on ratios from the prior knowledge of fluxes. Comprehensive assessments of methane sources and sinks have been provided by Saunois et al. (2016, 2020b) and Kirschke et al. (Kirschke et al., 2013).

EDGAR (Crippa et al., 2019, 2021; Janssens-Maenhout et al., 2019) is one of multiple global methane BU inventories available. Other inventories – namely GAINS (Höglund-Isaksson, 2012), US-EPA (EPA, 2011, 2021), CEDS (Hoesly et al., 2018; McDuffie et al., 2020; O'Rourke et al., 2020) as well as FAOSTAT-CH$_4$ (Federici et al., 2015; Tubiello, 2018; Tubiello et al., 2013) – can differ in terms of their country and sector coverage as well as detail. EDGAR, CEDS, US-EPA and GAINS cover all major source sectors (fossil fuels, agriculture and waste, biofuel) – except large scale biomass burning – but this can

be added from different databases such as FINN (Wiedinmyer et al., 2011), GFAS (Kaiser et al., 2012), GFED (van der Werf et al., 2017) or QFED (Darmenov and da Silva, 2013). Much like $CO_2$ FFI, these inventories of anthropogenic emissions are not completely independent as they either follow the same IPCC methodology to derive emissions, rely on similar data sources (e.g., FAOSTAT activity data for agriculture, reported fossil fuel production), or draw on reported country inventory data (Petrescu et al., 2020a, e.g. Figure 4). However, they may differ in the assumptions and data used for the calculation. While

the US-EPA inventory uses the reported emissions by the countries to UNFCCC, other inventories produce their own estimates using a consistent approach for all countries, and country specific activity data, emission factor and technological abatement when available. FAOSTAT and EDGAR mostly apply a Tier 1 approach to estimate CH$_4$ emissions while GAINS uses a Tier 2 approach (Höglund-Isaksson et al., 2020). CEDS is based on pre-existing emission estimates from FAOSTAT and EDGAR and then scales these emissions to match country-specific inventories, largely those reported to UNFCCC.


Global anthropogenic CH$_4$ emission estimates are compared in Figure 1. EDGARv5 has revised total global CH$_4$ emissions about 10 Mt CH$_4$ yr$^{-1}$ higher than EDGARv4.3.2 due to a higher estimate for the waste sector (see supplementary material). Subsequent revisions of the estimation methodology in EDGARv6 in alignment with the IPCC guidelines refinement (IPCC, 2019) lead to very substantial differences in total CH$_4$ emissions that are up to 50 MtCH$_4$yr$^{-1}$ lower before the 1990s compared

to EDGARv5 and EDGARv4.3.2, but differences are smaller ranging from 1-13 MtCH$_4$yr$^{-1}$ since the 2000s (see Figure SM-1). The cause of these differences is a new procedure to separately estimate of the venting component for gas and oil, in the venting and flaring sector (1B2a/b2). Differences across different versions of the EDGAR dataset are shown in the Supplementary Material (Fig. SM-1). US-EPA show the lowest estimates probably due to missing estimates from a significant

number of countries not reporting to UNFCCC (US-EPA2020 includes estimates from only 195 countries) and incomplete sectoral coverage. EDGARv6 estimates of anthropogenic $CH_4$ emissions, as used here, are in the upper range of the different inventories across most anthropogenic sources. However, none of these inventories cover $CH_4$ emissions from forest and grassland burning, which amount to about 10-12 Mt $yr^{-1}$.

Saunois et al (2020b) provide estimates of $CH_4$ sources and sinks based on BU and TD approaches associated with an uncertainty range based on the minimum and maximum values of available studies (because for many individual source and sink estimates the number of studies is often relatively small). Thus, they do not consider the uncertainty of the individual estimates. As shown in Table 5, uncertainties in total global $CH_4$ emissions across all anthropogenic and natural sources are comparatively small from TD approaches at ±6% - a range larger than errors in transport models only (Locatelli et al., 2015). However, this uncertainty on total emissions is probably underestimated as the uncertainty in the chemical sink was not fully considered in the TD estimates in Saunois et al (2020). Uncertainty on the global burden of OH is about ±5%, much lower than EDGARv4.3.2 and v5 uncertainties derived from the detailed analysis of (Janssens-Maenhout et al., 2019) and Solazzo et al. (2021), reaching around ±45% at 2σ. Saunois et al. (2020) reported uncertainty of 10-15%, which translates to an uncertainty of about ±10% to ±30% depending on the category, with larger uncertainty in the fossil fuel sectors than in the agriculture and waste sector (Saunois et al., 2020b). However, these uncertainties are also underestimated as they do not consider the uncertainty in each individual estimate, which includes potential uncertainties in activity data, emission factors, and equations used to estimate emissions.

Uncertainties in EDGAR $CH_4$ emissions using a Tier 1 approach are estimated at -33% to +46% at 2σ, but there is great variability across individual sectors ranging from ±30% (agriculture) to more than ±100% (fuel combustion), with high uncertainties in oil and gas sector (±93%) and coal fugitive emissions (±65%) (Solazzo et al., 2021). Inventories at national scale, such as in the USA also show large uncertainties depending on the sector (NASEM, 2018), though the activity data uncertainty may be lower than those for less developed countries. For example, global inventories, such as EDGAR, estimate uncertainties in national anthropogenic emissions of about ± 32% for the 24 member countries of OECD, and up to ±57% for other countries, whose activity data are more uncertain (Janssens-Maenhout et al., 2019).

**Table 5 - Uncertainties estimated for $CH_4$ sources at the global scale:** based on ensembles of bottom-up (BU) and top-down (TD) estimates, national reports and specific uncertainty assessments of EDGAR. Note that this tables provides uncertainty estimates from some of the key literature based on different methodological approaches. It is not intended to be an exhaustive treatment of the literature.

| | Estimated uncertainty in USA inventories [a] | Janssens-Maenhout et al. (2019) EDGARv4.3.2 uncertainty at 2σ | Solazzo et al. (2021) EDGARv5 uncertainty at 2σ | Global inventories uncertainty range [b] | Saunois et al. (2020) BU uncertainty range[c] | Saunois et al. (2020) TD uncertainty range[c] |
|---|---|---|---|---|---|---|
| Total global anthropogenic | | | | - | ±6% | ±6% |

| | | | | | | |
|---|---|---|---|---|---|---|
| sources (incl. Biomass burning) | | | | | | |
| Total global anthropogenic sources (excl. Biomass burning) | | ±47% | -33% to +46% | ±8% | ±5% | |
| Agriculture and Waste | | | | | ±8% | ±8% |
| Rice | na | ±60% | 31-38% | ±22% | ±20% | - |
| Enteric fermentation and manure management | ±10 to 20%<br>± 20% and up to ± 65% | | ±5% | ±8% | | - |
| Landfills and Waste | ±10% but likely much larger | ±91% | 78-79% | ±17% | ±7% | - |
| Fossil fuel production & use | | | | | ±20% | ±25% |
| Coal | -15% to +20% | ±75% | 65%  60-74% | ±40% | ±28% | - |
| Oil and gas | -20 % to +150% | 93% | ±19% | ±15% | | - |
| Other | na | ±100% | ±100% | ±64% | ±130%* | |
| Biomass and biofuel burning | | | | - | ±25% | ±25% |
| Biomass burning | | | | - | ±35% | |
| Biofuel burning | Included in "Other" | | 147% | +/-24% | ±17% | |

[a] Based on (NASEM, 2018)

[b] Uncertainty calculated as $((min-max)/2)/mean*100$ from the estimates of year 2017 of the six inventories plotted in Figure 1. This does not consider uncertainty on each individual estimate.

[c] Uncertainty calculated as $((min-max)/2)/mean*100$ from individual estimates for the 2008-2017 decade. This does not consider uncertainty on each individual estimate, which is probably larger than the range presented here.

[*] Mainly due to difficulties in attributing emissions to small specific emission sector.

The most recent UN emissions gap report (UNEP, 2020) gives an uncertainty range for global anthropogenic $CH_4$ emissions with one standard deviation of ±30% (i.e. ±60% for $2\sigma$), which is slightly higher than recent estimates in the literature. On the other hand, IPCC AR5 provides a comparatively low estimates at ±20% for a 90% confidence interval. Overall, we apply a best value judgment of ±30% for global anthropogenic $CH_4$ emissions for a 90% confidence interval. This is justified by the

larger uncertainties reported in uncertainties studies on the EDGAR dataset (Janssens-Maenhout et al., 2019; Solazzo et al., 2021) as well as for FAO activity statistics by Tubiello et al. (2015).

### 3.4 Anthropogenic $N_2O$ emissions

Anthropogenic $N_2O$ emissions occur in a number of sectors, namely agriculture, fossil fuel and industry, biomass burning, and waste. The emissions from the agriculture sector have four components: direct and indirect emissions from soil and water
bodies (inland, coastal, and oceanic waters), manure left on pasture, manure management, and aquaculture. Besides these main sectors, a final 'other' category represents the sum of the effects of climate, elevated atmospheric $CO_2$, and land cover change. This is a new sector that was developed as part of the global nitrous oxide budget (Tian et al., 2020) – a recent assessment to quantify all sources and sinks of $N_2O$ emissions, updating previous work (Kroeze et al., 1999; Mosier et al., 1998; Mosier and Kroeze, 2000; Syakila and Kroeze, 2011). Overall, anthropogenic sources contributed just over 40% to total global $N_2O$
emissions (Tian et al., 2020).

There are a variety of approaches for estimating $N_2O$ emissions. These include inventories (Janssens-Maenhout et al., 2019; Tian et al., 2018; Tubiello et al., 2013), statistical extrapolations of flux measurements (Wang et al., 2020), and process-based land and ocean modelling (Tian et al., 2019; Yang et al., 2020). There are at least five relevant global $N_2O$ emissions inventories
available: EDGAR (Crippa et al., 2019, 2021; Janssens-Maenhout et al., 2019), GAINS (Winiwarter et al., 2018), FAOSTAT-$N_2O$ (Tubiello, 2018; Tubiello et al., 2013), CEDS (Hoesly et al., 2018; McDuffie et al., 2020; O'Rourke et al., 2020) and GFED (van der Werf et al., 2017). While EDGAR and GAINS cover all sectors except biomass burning, FAOSTAT-$N_2O$ is focused on agriculture and biomass burning and GFED on biomass burning only. As shown in Figure 1 EDGAR, GAINS, CEDS and FAOSTAT emissions are consistent in magnitude and trend. Recent revisions in estimating indirect $N_2O$ emissions
in EDGARv6 lead to an average increase of 1.5% $yr^{-1}$ in total $N_2O$ emissions estimates between 1999 and 2018 compared to EDGARv5 (differences before 1999 were negligible at less than 1% $yr^{-1}$ ). Differences across different versions of the EDGAR dataset are shown in the Supplementary Material (Figure SM-1). The main discrepancies across different global inventories are in agriculture, where emission estimates from the global nitrous oxide budget (also referred to as "GCP") (Tian et al., 2020) and FAOSTAT are on average 1.5 Mt $N_2O$ $yr^{-1}$ higher than those from GAINS and EDGAR during 1990-2016, due to much
higher estimates of direct emissions from fertilised soils and manure left on pasture. GCP provides the largest estimate, because it synthesised from the other three inventories and further informed by additional bottom-up modelling estimates – and is as such more comprehensive in scope (Figure 1). In particular, it includes an additional sector that considers the sum of the effects of climate, elevated atmospheric $CO_2$, and land cover change (Tian et al., 2020). EDGAR estimates of anthropogenic $N_2O$ emissions as used in this dataset should therefore be considered as lower bound estimates. Differences in $N_2O$ emissions across
different versions of EDGAR are shown in Figure SM-1.

Anthropogenic $N_2O$ emissions estimates are subject to considerable uncertainty – larger than those from FFI-$CO_2$ or $CH_4$ emissions. $N_2O$ inventories suffer from high uncertainty on input data, including fertiliser use, livestock manure availability, storage and applications (Galloway et al., 2010; Steinfeld et al., 2010) as well as nutrient, crops and soils management (Ciais et al., 2014; Shcherbak et al., 2014). Emission factors are also uncertain (Crutzen et al., 2008; Hu et al., 2012; IPCC, 2019; Yuan et al., 2019) and there remains several sources that are not yet well understood (e.g. peatland degradation, permafrost) (Elberling et al., 2010; Wagner-Riddle et al., 2017; Winiwarter et al., 2018). Model-based estimates face uncertainties associated with the specific model configuration as well as parametrisation (Buitenhuis et al., 2018; Tian et al., 2018, 2019). Total uncertainty is also large because $N_2O$ emissions are dominated by emissions from soils, where our level of process understanding is rapidly changing.

For EDGARv4.3.2 uncertainties in $N_2O$ emissions are estimated based on default values (IPCC, 2006) at ±42% for 24 OECD90 countries and at ±93% for other countries for a 95% confidence interval (Janssens-Maenhout et al., 2019). However, Solazzo et al. (2021) arrive at substantially larger values for EDGAR allowing for correlation of uncertainties between sectors, countries and regions. At a sector level, uncertainties are larger for agriculture (263%) than for energy (113%), waste (181%), industrial processes and product use (14%) and other (112%). In the recent Emissions Gap Report (UNEP, 2020) relative uncertainties for global anthropogenic $N_2O$ emissions are estimated at ±50% for a 68% (1σ) confidence interval. This is larger than the ±60% uncertainties reported in IPCC AR5 for a 90% confidence interval (Blanco G. et al., 2014), but is comparable with the ranges for anthropogenic emissions in the global $N_2O$ budget (Tian et al., 2020). Overall, we assess the relative uncertainty for global anthropogenic $N_2O$ emissions at ±60% for a 90% confidence interval.

**Table 6 - Comparison of four global N$_2$O inventories:** EDGAR (Crippa et al., 2021); GCP (Tian et al., 2020); GAINS (Winiwarter et al., 2018); FAOSTAT (FAOSTAT, 2021; Tubiello, 2018; Tubiello et al., 2013)

| Name | Time coverage | Geographical coverage | | Activity split | IPCC emissions factors | Reported emissions in 2015 (in MtN$_2$O) | | | | | |
|------|------|------|------|------|------|------|------|------|------|------|------|
| | | | | | | agriculture | Fossil fuel and industry | Biomass burning | Waste and waste sector | other | Total |
| EDGAR | 1970-2018 | Global, countries | 228 | 4 main sectors, 24 sub-sectors | Yes | 6.2 | 2.3 | 0.05 | 0.4 | - | 8.9 |
| GCP | 1980-2016 | Global, regions | 10 | 5 main sectors, 14 sub-sectors | no | 8.4 | 1.6 | 1.1 | 0.6 | 0.3 | 11.9 |
| GAINS | 1990-2015 (every 5 years) | Global, regions | 172 | 3 main sectors, 16 sub-sectors | no | 6.8 | 1.3 | - | 0.7 | - | 8.8 |
| FAOSTAT | 1961-2019 | Global, countries | 231 | 2 main sectors, 9 sub-sectors | Yes | 8.3 | - | 0.9 | - | - | 9.2 |


### 3.5 Fluorinated gases

Fluorinated gases comprise over a dozen different species that are primarily used as refrigerants, solvents and aerosols. Here we compare global emissions of F-gases estimated in EDGARv5 and v6 to top-down estimates from the 2018 World Meteorological Organisation's (WMO) Scientific Assessment of Ozone Depletion (Engel and Rigby, 2018; Montzka and

Velders, 2018). We provide additional comparisons with other EDGAR versions as well as estimates by the US-EPA in the supplementary material (see Figure SM-2). The top-down estimates were based on measurements by the Advanced Global Atmospheric Gases Experiment (AGAGE, Prinn et al., 2018) and National Oceanic and Atmospheric Administration (NOAA, Montzka et al., 2015), assimilated into a global box model (using the method described in Engel and Rigby, et al., 2018 and Rigby et al., (2014)). Uncertainties in the top-down estimates are due to measurement and transport model uncertainty. As F-

gas emissions are almost entirely anthropogenic in nature, top-down estimates of anthropogenic fluxes are much better known than CO$_2$, CH$_4$, N$_2$O, where large natural fluxes contribute to the observed trends. For substances with relatively short lifetimes (~50 years or less), uncertainties are typically dominated by uncertainties in the atmospheric lifetimes. Comparisons between the EDGARv5, v6 and WMO 2018 estimates were available for HFCs 125, 134a, 143a, 152a, 227ea, 23, 236fa, 245fa, 32, 365mfc and 43-10-mee, PFCs CF$_4$, C$_2$F$_6$, C$_3$F$_8$ and c-C$_4$F$_8$, SF$_6$ and NF$_3$ (EDGAR v6 only). For the higher molecular weight

PFCs ($C_4F_{10}$, $C_5F_{12}$, $C_6F_{14}$, $C_7F_{16}$), top-down estimates were not available in WMO (2018). Top-down estimates have previously been published for these compounds (e.g. Ivy et al., 2012), however, this comparison is not included here due to their very low emissions. For a small number of species, global top-down estimates are available for some years, based on an independent atmospheric model to that used in WMO (2018), although most of these inversions use similar measurement datasets; Fortems-Cheiney, et al. (2015) for HFC-134a, Lunt, et al. (2015) for HFC-134a, -125, -152a, -143a and -32 and

Rigby, et al (2010) for $SF_6$.

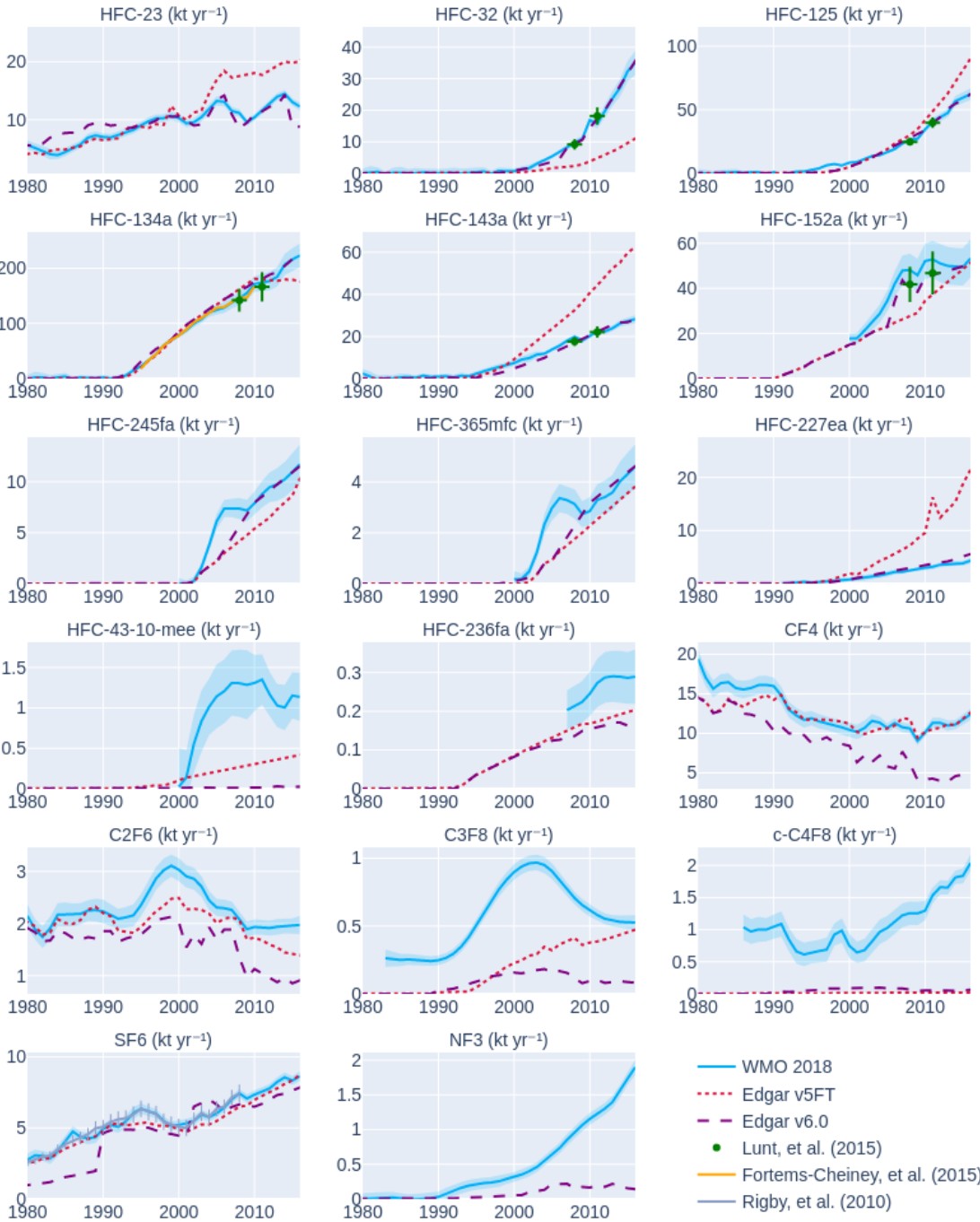

**Figure 2 - Comparison of top-down and bottom-up estimates for individual species of fluorinated gases in Olivier and Peters (2020) [EDGARv5FT] and EDGARv6 for 1980-2016.** $C_4F_{10}$, $C_5F_{12}$, $C_6F_{14}$ and $C_7F_{16}$ are excluded. Top-down estimates from WMO 2018 (Engel and Rigby, 2018; Montzka and Velders, 2018) are shown as blue lines with blue shading indicating $1\sigma$ uncertainties. Bottom-up estimates from EDGARv5 and v6 are shown in red dotted lines and purple dashed lines, respectively. Top-down estimates for some species are shown from Rigby, et al. (2010), Lunt, et al. (2015) and Fortems-Cheiney, et al. (2015).

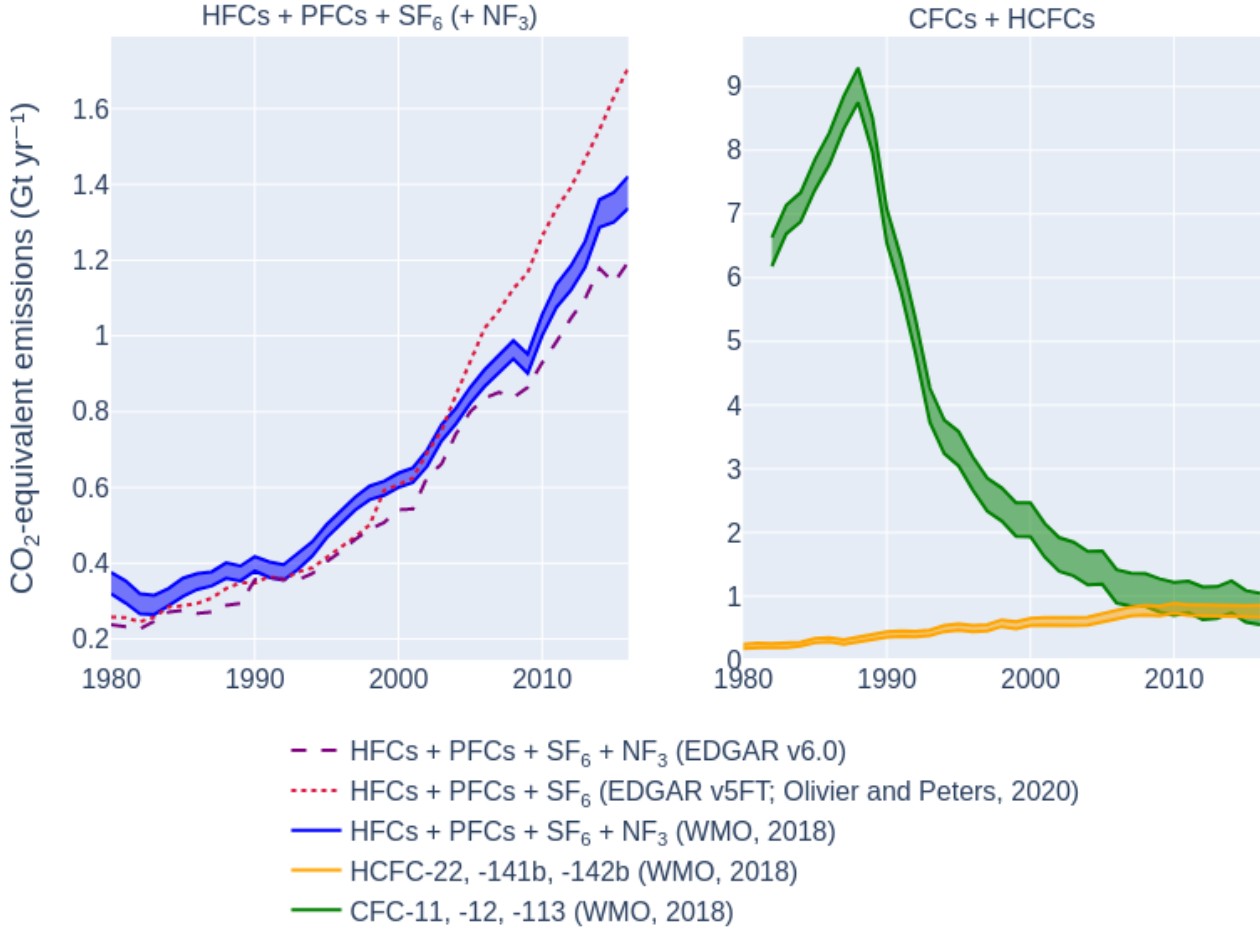

**Figure 3 – Comparison between top-down estimates and bottom-up EDGARv5 and v6 inventory data on GHG emissions for 1980-2016.** Left panel: Total GWP-100-weighted emissions based on IPCC AR6 (Forster et al., 2021) of F-gases in Olivier & Peters (2020) [EDGARv5FT] (red dashed line, excluding $C_4F_{10}$, $C_5F_{12}$, $C_6F_{14}$ and $C_7F_{16}$) and EDGARv6 (purple dashed line) compared to top-down estimates based on AGAGE and NOAA data from WMO (2018) (blue lines; Engel and Rigby, (2018); Montzka and Velders (2018)). Right panel: Top-down aggregated emissions for the three most abundant CFCs (-11, -12 and -113) and HCFCs (-22, -141b, -142b) not covered in bottom-up emissions inventories are shown in green and orange. For top-down estimates the shaded areas between two respective lines represent $1\sigma$ uncertainties.

The comparison of global top-down and bottom-up emissions for EDGARv6 and Olivier & Peters 2020 [EDGARv5FT] F-gas species (excluding heavy PFCs) is shown in Figure 2 for the years 1980 – 2016 (or a subset thereof, depending on the availability of the top-down estimates). Where available, the various top-down estimates agree with each other within

uncertainties. The magnitude of the difference between the WMO (2018) and EDGAR estimates varies markedly between species, years and versions of EDGAR; for several HFCs, the top-down and bottom-up estimates often agree within uncertainties for EDGARv6 (but much less often in v5), whereas for c-$C_4F_8$, the top-down estimate is more than 100 times the EDGAR estimates. Some similarities and differences have been previously noted for earlier versions of EDGAR (Lunt et al., 2015; Mühle et al., 2010, 2019; Rigby et al., 2010). For $SF_6$, the relatively close agreement between EDGAR v4.0 and a top-down estimate has been discussed in Rigby, et al. (2010). They estimated uncertainties in EDGAR v4.0 of ±10% to ±15%, depending on the year, and indeed, top-down values were consistent within these uncertainties. However, the agreement is now poorer during the 1980s in EDGARv6. For some PFCs (e.g., $CF_4$, $C_2F_6$), it was previously noted that some assumptions within EDGAR v4.0 had been validated against atmospheric observations, hence EDGAR might be considered a hybrid of top-down and bottom-up methodologies for these species (Mühle et al., 2010). However, it is unclear for which other species similar validation has taken place, or how these assumptions vary between versions of EDGAR.

When species are aggregated into F-gas total emissions, weighted by their current 100-year GWPs based on IPCC AR6 (Forster et al., 2021), we note that in the left panel of Figure 3 the Olivier & Peters (2020) [EDGARv5FT] estimates are around 10% lower than the WMO 2018 values in the 1980s. Subsequently, EDGARv5FT estimates grow more rapidly than the top-down values and are almost 30% higher than WMO 2018 by the 2010s. EDGARv6 emissions are around 10% lower than the WMO 2018 values throughout. Given that detailed uncertainty estimates are not available for all EDGAR F-gas species, we base our uncertainty estimate solely on this comparison with the top-down values (see Figure 3, left panel), and therefore suggest a conservative uncertainty in aggregated F-gas emissions of ±30% for a 90% confidence interval. For individual species, the magnitude of this discrepancy can be orders of magnitude larger.

The F-gases in EDGAR do not include chlorofluorocarbons (CFCs) and hydrochlorofluorocarbons (HCFCs), which are groups of substances regulated under the Montreal Protocol. Historically, total $CO_2eq$ F-gas emissions have been dominated by the CFCs (Engel and Rigby, 2018). In particular, during the 1980s, peak annual emissions due to CFCs reached 9.1±0.4 $GtCO_2eqyr^{-1}$ (Figure 3), comparable to that of $CH_4$, and substantially larger than the 2018 emissions of the gases included in EDGARv5 and v6 (1.3 $GtCO_2eq$) (Table 7). Subsequently, following the controls of the Montreal Protocol, emissions of CFCs declined substantially, while those of HCFCs and HFCs rose, such that $CO_2eq$ emissions of the HFCs, HCFCs and CFCs were approximately equal by 2016, with a smaller contribution from PFCs, $SF_6$, $NF_3$ and some more minor F-gases. Therefore, the GWP-weighted F-gas emissions in EDGAR, which are dominated by the HFCs, represent less than half of the overall $CO_2eq$ F-gas emissions in 2016.

**3.6 Aggregated GHG emissions**

Based on our assessment of relevant uncertainties above, we apply constant, relative uncertainty estimates for GHGs at a 90% confidence interval that range from relatively low for $CO_2$ FFI (±8%), to intermediate values for $CH_4$ and F-gases (±30%), to

higher values for $N_2O$ (±60%) and $CO_2$ from LULUCF (±70%). To aggregate these and estimate uncertainties for total GHGs in terms of $CO_2$eq emissions, we are taking the square root of the squared sums of absolute uncertainties for individual (groups of) gases, using 100-year Global Warming Potentials (GWP-100) with values from IPCC AR6 (Forster et al., 2021, Section 7.6 and Supplementary Material 7.SM.6) to weight emissions of non-$CO_2$ gases but excluding uncertainties in the metric itself (see Section 3.7). Overall, this is broadly in line with IPCC AR5 (Blanco G. et al., 2014), but provides important adjustments in the evaluation of uncertainties of individual gases ($CH_4$, F-gases, $CO_2$-LULUCF) as well as the approach in reporting total uncertainties across GHGs.

## 3.7 GHG emission metrics

GHG emission metrics are necessary if emissions of non-$CO_2$ gases and $CO_2$ are to be aggregated into $CO_2$eq emissions. GWP-100 is the most common metric and has been adopted for emissions reporting under the transparency framework for the Paris Agreement (UNFCCC, 2019), but many alternative metrics exist in the scientific literature. The most appropriate choice of metric depends on the climate policy objective and the specific use of the metric to support that objective (i.e. why do we want to aggregate or compare emissions of different gases? What specific actions do we wish to inform?)

Different metric choices and time horizons can result in very different weightings of the emissions of Short-lived Climate Forcers (SLCF), such as $CH_4$. For example, 1t $CH_4$ represents as much as 81 t$CO_2$eq if a Global Warming Potential is used with a time horizon of 20 years, or as little as 5.4t $CO_2$eq if the Global Temperature change Potential (GTP) is used with a time horizon of 100 years (Forster et al., 2021). More recent metric developments that compare emissions in new ways – e.g. the additional warming from sustained changes in SLCF emissions compared to pulse emissions of $CO_2$ – increase the range of metric values further and can even result in negative metric values for SLCF, if their emissions are falling rapidly (Allen et al., 2018; Cain et al., 2019; Collins et al., 2019; Lynch et al., 2020).

The contribution of SLCF emissions to total GHG emissions expressed in $CO_2$eq thus depends critically on the choice of GHG metric and its time horizon. However, even for a given choice, the metric value for each gas is also subject to uncertainties. For example, the GWP-100 for biogenic $CH_4$ has changed from 21 based on the IPCC Second Assessment Report (SAR) in 1995 to 28 or 34 based on the IPCC AR5 (excluding or including climate-carbon cycle feedbacks), and to 27 based on IPCC AR6. These changes and remaining uncertainties arise from parametric uncertainties, differences in methodological choices, and changes in metric values over time due to changing background conditions.

- Parametric uncertainties arise from uncertainties in climate sensitivity, radiative efficacy and atmospheric lifetimes of $CO_2$ and non-$CO_2$ gases, etc. The IPCC AR6 assessed the parametric uncertainty of GWP for $CH_4$ as ±32% and ±40% for time horizons of 20 and 100 years, ±43% and ±47% for $N_2O$, and ±26-31 and ±33-38% for various F-gases

(Forster et al., 2021). The uncertainty of GTP-100 for $CH_4$ was estimated at ±83%, which is larger than the uncertainty in a forcing-based metric due to due to uncertainties in climate responses to forcing (e.g., transient climate sensitivity).

- Methodological choices introduce a different type of uncertainty, namely which indirect effects are included in the calculation of metric values and the strength of those feedbacks. For $CH_4$, indirect forcing caused by photochemical decay products (mainly tropospheric ozone and stratospheric water vapour) contributes almost 40% of the total
705 forcing from $CH_4$ emissions. More than half of the changes in GWP-100 values for $CH_4$ in successive IPCC assessments from 1995 to 2013 are due to re-evaluations of these indirect forcings. These uncertainties are incorporated in the above uncertainty estimates. In addition, warming due to the emission of non-$CO_2$ gases extends the lifetime of $CO_2$ already in the atmosphere through climate-carbon cycle feedbacks (Friedlingstein et al., 2013). Including these feedbacks results in higher metric values for all non-$CO_2$ gases, but the magnitude of this effect is
710 uncertain; e.g. the IPCC AR5 found the GWP-100 value for $CH_4$ without climate-carbon cycle feedbacks to be 28, whereas including this feedback would raise the value to between 31 and 34 (Gasser et al., 2016; Myhre et al., 2013; Sterner and Johansson, 2017). The IPCC AR6 decided to include clinate-carbon cycle feedbacks by default and no longer reports values without climate-carbon cycle feedbacks (Forster et al., 2021).

- A third uncertainty arises from changes in metric values over time. Metric values depend on the radiative efficacy of
715 $CO_2$ and non-$CO_2$ emissions, which in turn depend on the changing atmospheric background concentrations of those gases. Rising temperature can further affect the lifetime of some gases and hence their contribution to forcing over time for different emission scenarios (Reisinger et al., 2011). Successive IPCC assessments take changing starting-year background conditions into account, which explains part of the changes in GWP-100 metric values in different reports. Applying a single metric value to a multi-decadal historical time series of emissions is therefore only an
720 approximation of the correct metric value for any given emissions year, as e.g. the correct GWP-100 value for $CH_4$ emitted in the year 1970 will be different to the GWP-100 value for an emission in the year 2018. However, the literature does not offer a complete set of GWP-100 metric values for past concentrations and climate conditions covered in our time series.

Overall, we estimate the uncertainty in GWP-100 metric values, if applied to an extended historical emission time series, as ±50% for $CH_4$ and other SLCFs, and ±40% for non-$CO_2$ gases with longer atmospheric lifetimes (specifically, those with lifetimes longer than 20 years). If uncertainties in GHG metrics are considered, the overall uncertainty of total GHG emissions in 2018 increases from ±11% to ±24%. (However, in the following sections we do not include GWP uncertainties in our global, regional or sectoral estimates).

For the purpose of this paper, we use GWP-100 metric values from the IPCC AR5 (Myhre et al., 2013) without climate-carbon cycle feedbacks. Even though climate-carbon cycle feedbacks are considered a robust feature of the climate system, the issue was only emerging during the IPCC AR5 and the methodology used to include this in metric calculations was indicative only.

Subsequent studies (Gasser et al., 2016; Sterner and Johansson, 2017) suggest that revisions to the simple estimation method in IPCC AR5 are necessary.

As mentioned above, the most appropriate metric to aggregate GHG emissions depends on the objective. One such objective can be to understand the contribution of emissions in any given year to warming, while another can be to understand the contribution of cumulative emissions over an extended time period to additional warming relative to a given reference level. Sustained emissions of SLCFs such as $CH_4$ do not cause the same temperature response as sustained emissions of $CO_2$. Showing superimposed emission trends of different gases over multiple decades using GWP-100 as equivalence metric therefore does not necessarily represent the overall contribution to warming from each gas over that period. In Figure 4 we therefore also show the modelled warming from emissions of each gas or group of gases - calculated using the simple climate model emulator FaIRv1.6.2 and calibrated to reproduce the pulse-response functions for each gas consistent with the IPCC AR6 (see Forster et al., 2021, Supplementary Material 7.SM.3). There are some differences compared to the contribution of each gas, based on GHG emissions expressed in $CO_2$eq using GWP-100 (see Figure 8), in particular a greater contribution from $CH_4$ emissions to historical warming. This is consistent with warming from $CH_4$ being short-lived and hence having a more pronounced effect in the near-term during a period of rising emissions. Nonetheless, Figure 4 highlights that weighting emissions based on GWP-100 does not provide a vastly different overall story than modelled warming over the historical period when emissions of all gases have been rising, with $CO_2$ being the dominant and $CH_4$ being the second most important contributor to GHG-induced warming. Other metrics such as GWP* (Cain et al., 2019) offer an even closer resemblance between cumulative $CO_2$eq emissions and temperature change, relative to a specified starting point, especially if SLCF emissions are no longer rising but potentially falling, as in mitigation scenarios.

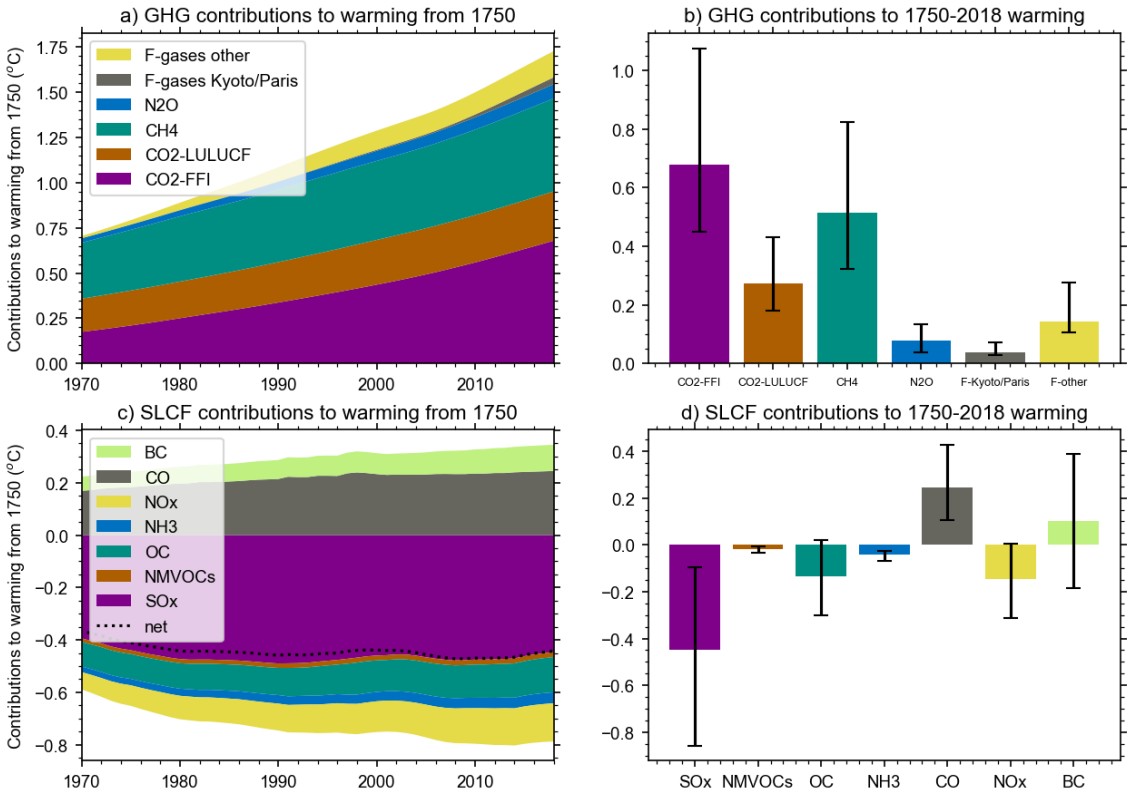

**Figure 4 - Contribution of different GHGs to global warming over the period 1750 to 2018.** Top row: contributions from estimated with the FaIR reduced-complexity climate model. Major GHGs and aggregates of minor gases as a timeseries in a) and as a total warming bar chart with 90% confidence interval added in b). Bottom row: contribution from short-lived climate forcers as a timeseries in c) and as a total warming bar chart with 90% confidence interval added in d). The dotted line in c) gives the net temperature change from short-lived climate forcers. F-Kyoto/Paris includes the gases covered by the Kyoto Protocol and Paris Agreement as well as the HFCs, while F-other includes 760 the gases covered by the Montreal Protocol but excluding the HFCs.

## 4 Results

Here we analyse global trends in anthropogenic GHG emissions in four time periods: (1) 1970-2018 to characterise the main trends in the data; (2) 2009-2018 to focus on the last decade; as well as (3) 2018 and (4) 2019 emission levels.

### 4.1 Global anthropogenic GHG emissions for 1970-2018

There is high confidence that global GHG emissions have increased every decade from an average of $31\pm4.3$ GtCO$_2$eqyr$^{-1}$ for the decade of the 1970s to an average of $55\pm5.9$ GtCO$_2$eqyr$^{-1}$ during 2009-2018 as shown in Table 7. The decadal growth rate

initially decreased from 1.7% yr$^{-1}$ in the 1970s (1970-1979) to 0.9% yr$^{-1}$ in the 1990s (1990-1999). After a period of accelerated growth during the 2000s (2000-2009) at 2.5%yr$^{-1}$, triggered mainly by growth in $CO_2$-FFI emissions from rapid industrialisation in China (Chang and Lahr, 2016; Minx et al., 2011), relative growth has decreased again to 1.2% yr$^{-1}$ during the most recent decade (2009-2018). Uncertainties in aggregate GHG emissions have decreased over time as the share of less uncertain $CO_2$-FFI emission estimates increased and the share of more uncertain emission estimates such as $CO_2$-LULUCF or $N_2O$ decreased.

**Table 7 – Average annual anthropogenic GHG emissions by decade and for selected individual years 1970-2018:** $CO_2$ from fossil fuel combustion and industrial processes (FFI); $CO_2$ from land use, land-use change and forestry (LULUCF); $CH_4$; $N_2O$; fluorinated gases (F-gases). Aggregate GHG emission trends by groups of gases reported in GtCO2eq converted based on global warming potentials with a 100-year time horizon (GWP-100) from the IPCC AR6 (Forster et al., 2021). Uncertainties are reported for a 90 % confidence interval (see Section 3). Levels and growth are average values over the indicated time period. Additional supplementary tables show similar average annual GHG emissions by decade also for major sectors (Table SM-2) and regions (Table SM-2).

| | Average annual emissions levels (GtCO2eq yr$^{-1}$) and average annual emissions growth (%) | | | | | | | | | | | |
| | CO$_2$ FFI | | CO$_2$ LULUCF | | CH$_4$ | | N$_2$O | | Fluorinated gases | | GHG | |
| | Levels | Growth | Levels | Growth | Levels | Growth | Levels | Growth | Levels | Growth | Levels | Growth |
|---|---|---|---|---|---|---|---|---|---|---|---|---|
| 2018 | 38±3.0 | | 5.7±4.0 | | 10±3.1 | | 2.5±1.5 | | 1.3±0.40 | | 58±6.1 | |
| 2009-2018 | 36±2.9 | 1.3% | 5.7±4.0 | 0.7% | 10±3.0 | 1.0% | 2.4±1.4 | 1.3% | 1.1±0.34 | 4.4% | 55±5.9 | 1.2% |
| 2000-2009 | 29±2.3 | 3.3% | 5.2±3.7 | -0.3% | 8.9±2.7 | 1.7% | 2.1±1.3 | 1.4% | 0.8±0.24 | 4.1% | 46±5.3 | 2.5% |
| 1990-1999 | 24±1.9 | 1.2% | 5.0±3.5 | -0.1% | 8.2±2.5 | 0.3% | 2.0±1.2 | 0.9% | 0.49±0.15 | 5.9% | 39±4.8 | 0.9% |
| 1980-1989 | 21±1.6 | 1.6% | 4.7±3.3 | 1.8% | 7.6±2.3 | 1.0% | 1.9±1.1 | 0.7% | 0.27±0.08 | 3.1% | 35±4.5 | 1.5% |
| 1970-1979 | 18±1.4 | 2.8% | 4.6±3.2 | -1.6% | 7.1±2.1 | 1.2% | 1.7±1.0 | 2.0% | 0.19±0.057 | 5.4% | 31±4.3 | 1.7% |
| 1970 | 16±1.3 | | 5.0±3.5 | | 6.7±2.0 | | 1.5±0.92 | | 0.14±0.043 | | 29±4.3 | |

There is high confidence that emission growth has been varied, but persistent across different groups of gases. Decade-by-decade increases in global average annual emissions have been observed consistently across all (groups of) GHGs (Table 7). $CO_2$-LULUCF emissions have been more stable compared to other GHGs, albeit uncertain, and only recently started to show an upward trend. The pace and scale of emission growth has varied across groups of gases. While average annual emissions of all GHGs together grew by about 75% from 31±4.3 GtCO$_2$eqyr$^{-1}$ during the 1970s (1970-1979) to 55±5.9 GtCO$_2$eqyr$^{-1}$ during the most recent decade (2009-2018), $CO_2$-FFI emissions doubled from 18±1.4 to 36±2.9 GtCO$_2$eqyr$^{-1}$ and F-gases grew almost fivefold from 0.19±0.057 to 1.1±0.34 GtCO$_2$eqyr$^{-1}$ across the same time period. In fact, persistent and fast growth in F-gas emissions has resulted in emissions levels that are now tracking at about 1.3±0.40 Gt CO$_2$eqyr$^{-1}$ in 2018 – 2.3% of total

GHG emissions measured as GWP-100. Increases in average annual emissions levels from the 1970s (1970-1979) to the most
recent decade (2009-2018) were lower for $CO_2$-LULUCF (22%), $CH_4$ (41%) as well as $N_2O$ (42%) (see Table 7).

However, there is low confidence that the reported increases in $CO_2$-LULUCF emissions by decade actually constitute a statistically robust trend given the large uncertainties involved. In fact, two bookkeeping models underlying the AFOLU data show opposing positive and negative trends (BLUE, H&N, respectively), while the third model (OSCAR), averaging over simulations that use either the same land-use forcing as BLUE (LUH2) or H&N (FAO/FRA), tracks the approximate mean of these (see also Section 3.2). Dynamic global vegetation models, which also use the LUH2 forcing, show higher estimates recently, explained by them considering the loss in sink capacity, while the bookkeeping models do not (see Figure 1). Overall, the different lines of evidence are inconclusive with regard to an upward trend in $CO_2$-LULUCF emissions.

Global anthropogenic GHG emissions grew continuously slower than world GDP across all (groups of) individual gases resulting in a sustained decline in the GHG intensity of GDP as shown in Figure 5. The only exception is the group of F-gases for which the GHG intensity of GDP has significantly increased since 1970, with a marked acceleration during the 1990s and the early 2000s, an intermediate drop in the late 2000s and a continued growth thereafter. Per capita GHG emissions have been fluctuating substantially, with a sustained decline in global per capita GHG emissions since the 1970s followed by an approximate 15-year period of continued growth from the 2000s. In recent years, per capita GHG emissions levels have stabilized without clear evidence for peaking. For $CO_2$-FFI emissions, sustained growth in per capita emissions can be observed since the mid-1990s levelling off during the last decade. Per capita emissions for $CO_2$-LULUCF, $CH_4$ and $N_2O$ declined consistently since the 1970, but this trend has flattened out since the mid-1990s or early 2000s. Per-capita F-gas emissions show sustained and rapid growth over the full time period, interrupted only by a small decline in the late 2000s.

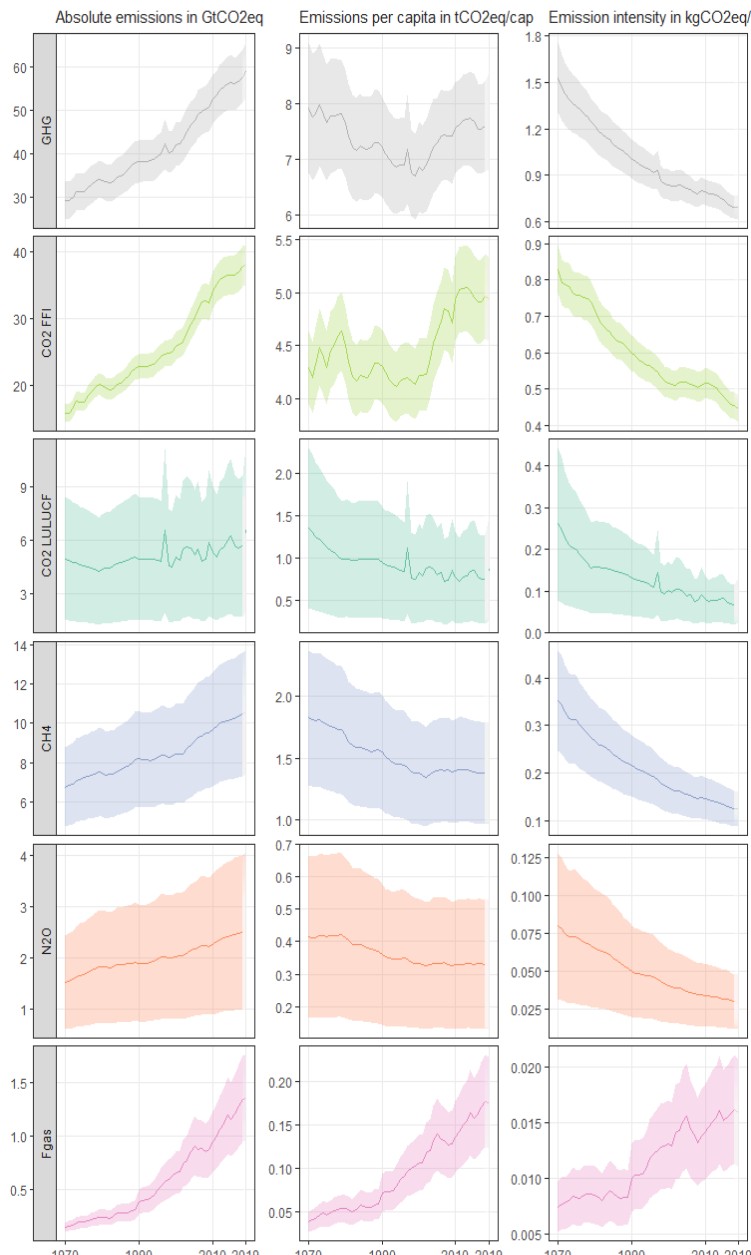

**Figure 5 - Global GHG emissions trends 1970-2019 by individual (groups of) gases and in aggregate:** GHGs (black); CO$_2$-FFI (light green); CO$_2$-LULUCF (dark green); CH$_4$ (blue); N$_2$O (orange); fluorinated gases (pink). Aggregate GHG emission trends by groups of gases reported in GtCO$_2$eq converted based on global warming potentials with a 100-year time horizon (GWP-100) from the IPCC AR6 (Forster et al., 2021). Coloured shadings show the associated uncertainties at a 90 % confidence interval without considering uncertainties in GDP and population data (see below). First column shows emission trends in absolute levels (GtCO$_2$eq). Second column shows per capita emissions trends (tCO$_2$eq/cap) using UN population data for normalization (World Bank, 2021). Third column shows emissions trends per unit of GDP (kgCO$_2$eq/$) using GDP data in constant 2010 $ from the World Bank for normalization (World Bank, 2021).

The continuous growth in global anthropogenic GHG emissions since the 1970s was mainly driven by activity growth in three major sectors: energy supply, industry and transportation (see Table SM-2; Fig. SM-4). In energy supply and transportation, average annual emissions were about 2.3 and 2.2 times larger for 2009-2018 than for 1970-1979, respectively, growing from 8.4 to 19 $GtCO_2eqyr^{-1}$ and 3.6 to 8.0 $GtCO_2eqyr^{-1}$, respectively. In industry, average annual GHG emissions were 1.8 times larger, growing from 7.3 $GtCO_2eqyr^{-1}$ in 1970-1979 to 13 $GtCO_2eqyr^{-1}$ in 2009-2018. At the sub-sector level, electricity & heat and road transport are the largest segments, growing 2.9 and 2.6 times between 1970-1979 and 2009-2018, respectively, from an average 4.6 to 13 $GtCO_2eqyr^{-1}$, and 2.2 to 5.7 $GtCO_2eqyr^{-1}$. The fastest growing sub-sector has been process emissions from cement, which is 4.1 times larger in 2009-2018 compared to 1970-1979, and currently accounts for an average 1.4 $GtCO_2eqyr^{-1}$. Other rapidly expanding sectors are international aviation (2.8 times larger on 1970-1979 levels), chemicals (1.9 times larger), metals (1.7 times larger), and waste (1.7 times larger). Growth in GHG emissions in AFOLU and buildings has been much more moderate with average annual GHG emissions only about 25% and 10% higher for 2009-2018 than for 1970-1979.

Most GHG emissions growth occurred in Asia and Developing Pacific as well as the Middle East, where emissions more than tripled from 6.3 $GtCO_2eqyr^{-1}$ and 0.8 $GtCO_2eqyr^{-1}$ in 1970-1979 to 23 $GtCO_2eqyr^{-1}$ and 2.8 $GtCO_2eqyr^{-1}$ in 2009-2018, respectively (see Table SM-1). Over the same time period GHG emissions grew 2.2 times in Africa and 1.7 times in Latin America and the Caribbean, while average annual anthropogenic GHG emissions levels in developed countries and Eastern Europe and West-Central Asia remained stable. However, Figure 6 highlights important variability at the country level. Note that these country-level estimates exclude $CO_2$-LULUCF emissions, because we assign low confidence to them. First, GHG emissions growth is taking place against the background of large differences in per capita GHG emissions between and within regions. For example, GHG emissions in developed countries have stabilized at high levels of per capita emissions compared to most other regions. Similarly, some countries in the Middle East are among the largest GHG emitters in per capita terms, while other countries of the region such as Yemen have seen comparatively little economic development showing low levels of per capita emissions. Second, the growth in GHG emissions has also been highly varied. For example, several developed countries in Europe such as UK, Germany or France have lower GHG emissions levels today than in the 1970s. In other countries like the US GHG emission levels are still considerably higher today even though they have recently started reducing GHG emissions – unlike Australia or Canada, which have until now only begun stabilizing their GHG emission levels. A comprehensive assessment of country progress in reducing GHG emissions can be found in Lamb et al. (2021b).

### a. GHG emissions per capita in 2018 in tCO2eq

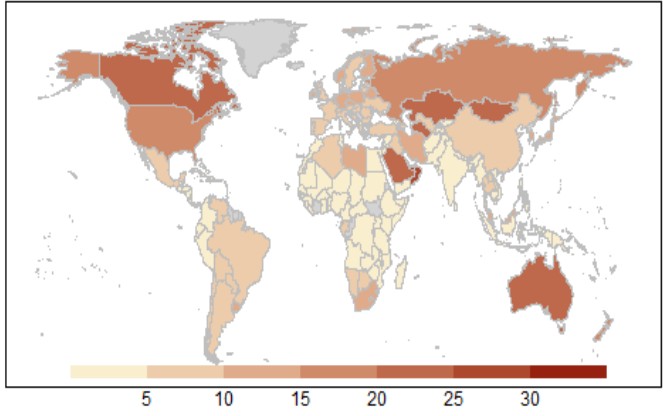

### b. Average annual GHG emissions growth in percent (2009-2018)

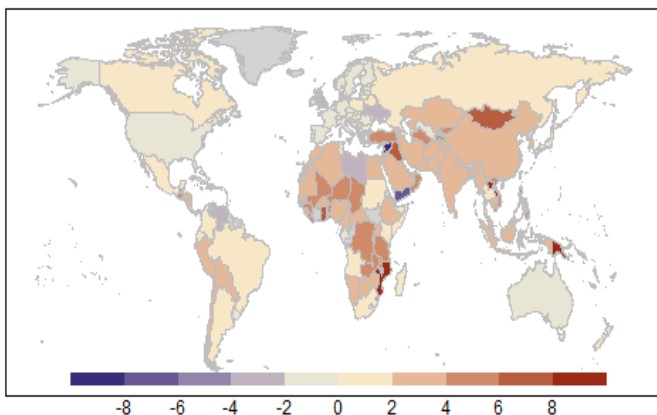

### c. Average annual GHG emissions growth in percent (1970-2018)

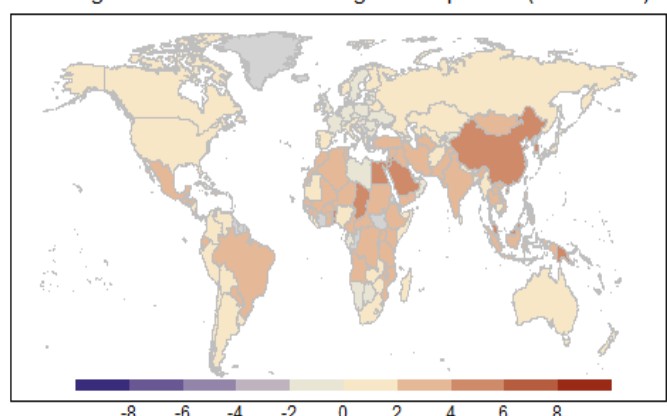

**Figure 6 – Levels of and changes in GHG emissions by country.** Aggregate GHG emissions are reported in $GtCO_2eq$ based on global warming potentials with a 100-year time horizon (GWP-100) from IPCC AR6 (Forster et al., 2021). Panel a shows per capita GHG emission levels ($tCO_2eqyr^{-1}$) for the year 2018 using UN population data for normalization (World Bank, 2021). Panel b shows average annual changes (in %) in GHG emissions by countries for 2009-2018. Panel c shows average annual changes (in %) in GHG emissions by countries for 1970-2018. Note that this excludes $CO_2$-LULUCF as there is currently low confidence in national level estimates.

In Figure 7 we compare historic GHG emission trends with different scenarios, to explore how emissions are developing relative to the range of projected future outcomes. The Integrated Assessment Modelling (IAM) community quantified five

Shared Socioeconomic Pathways (SSPs) for different levels of radiative forcing in 2100 using six different IAMs (Riahi et al., 2017; Rogelj et al., 2018b). The SSPs are grouped according to their radiative forcing ranging from 1.9 $Wm^{-2}$ to 8.5 $Wm^{-2}$, aimed at spanning the full range of potential outcomes. The Coupled Model Intercomparison Project Phase 6 (CMIP6) (Eyring et al., 2016) took a subset of these quantified SSPs as the basis for future climate projections (Gidden et al., 2019; O'Neill et al., 2016). In recent years, the use of the very high forcing scenarios – particularly SSP5-8.5 - is being debated in the scientific

community (e.g. Hausfather and Peters, 2020b, 2020a; Pedersen et al., 2020; Schwalm et al., 2020).

Historical GHG emissions from our database are consistent with the levels and trends in the scenario data, despite the scenarios being calibrated on older data sources (Gidden et al., 2019) – mainly CEDS (Hoesly et al., 2018). The observed differences are larger for the GHGs with the highest uncertainty, notably $CO_2$-AFOLU, $N_2O$ and F-gas emissions (Sections 3.2, 3.4 and 3.5). Across the different GHGs, historical emissions track on aggregate with the higher forcing scenarios such as the SSP3-

7.0 and SSP5-8.5 markers, in terms of both levels and growth rates. $CO_2$-FFI emissions still tend towards the higher end of the scenario range shown here, but there are signs that $CO_2$-FFI emissions are slowing to more moderate forcing levels (e.g., SSP4-6.0 and SSP2-4.5) when considering recent trends (Hausfather and Peters, 2020a). $CH_4$ and $N_2O$ emissions sit more in the middle and at the lower-end of the scenario range – the latter driven by the lower levels of $N_2O$ emissions in EDGAR – and F-gases are consistent with the scenarios. Total GHG emissions track the higher end scenarios.


Figure 7 highlights the very different future emission trajectories envisioned by IAMs for individual gases – particularly at radiative forcing levels that are consistent with the goal of the Paris Agreement such as SSP1-2.6 and SSP1-1.9. In contrast to $CO_2$ emission, non-$CO_2$ forcers such as anthropogenic $CH_4$ and $N_2O$ emissions are not reduced to zero. However, in many scenarios, F-gases reach zero emissions. $N_2O$ emissions remain at similar levels to today in some of the scenarios with a 1.9

$Wm^{-2}$ forcing at the end of the century, while they are about halved in others. Reductions in $CH_4$ emissions are a bit more pronounced ranging from about 100 to 200 $MtCH_4yr^{-1}$ in 2100 compared to almost 400 $MtCH_4yr^{-1}$ in 2019. $CO_2$-AFOLU emission trajectories overlap for different forcing levels, partly reflecting the complexities of modelling land-use change, but overall show a tendency towards a net carbon sink even in SSPs with little or even without climate policy. Given recent trends in land-use change emissions, it could be questioned whether the scenarios adequately explore the uncertainty in future land-

use change emissions (Hausfather and Peters, 2020b).

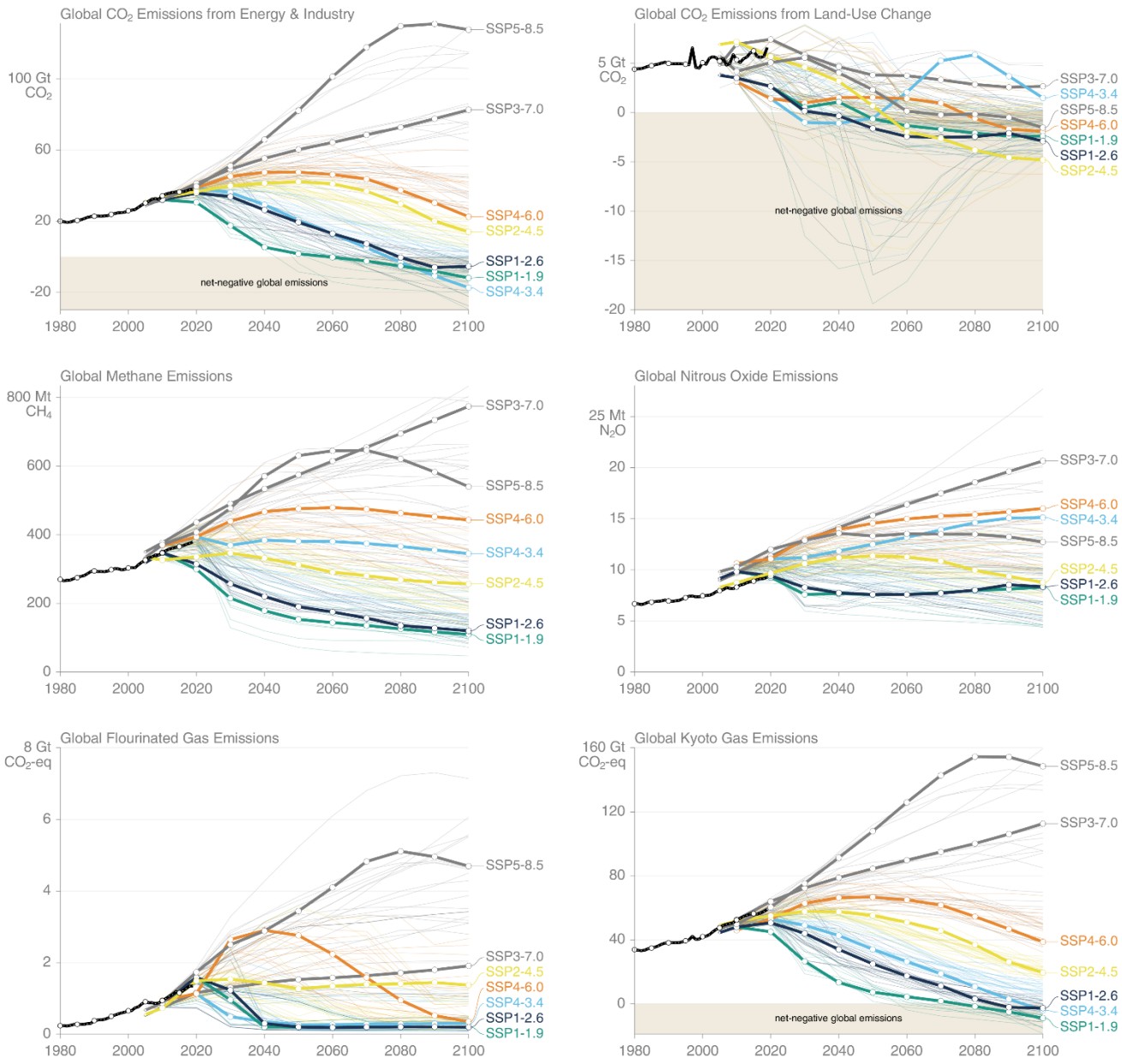

**Figure 7 - Historical emissions of GHGs and future projections in socio-economic scenarios.** The historical emissions are from this dataset. GHG emissions are reported in GtCO2eq converted based on global warming potentials with a 100-year time horizon (GWP-100) from the IPCC AR6 (Forster et al., 2021). The Shared Socioeconomic Pathways (SSPs) are from the SSP database version 2 (Riahi et al., 2017; Rogelj et al., 2018b). See also: https://tntcat.iiasa.ac.at/SspDb/). Highlighted scenarios are the markers used in CMIP6 (O'Neill et al., 2016) after harmonisation (Gidden et al., 2019).

## 4. 2 - Global GHG emissions for the last decade 2009-2018

There is high confidence that global anthropogenic GHG emission levels were higher in 2009-2018 than in any previous decade and GHG emission levels have grown across the most recent decade. Average annual GHG emissions for 2009-2018 were $55\pm5.9$ GtCO$_2$eqyr$^{-1}$ compared to $46\pm5.3$ and $39\pm4.8$ GtCO$_2$eqyr$^{-1}$ for 2000-2009 and 1990-1999, respectively. Marking an increase of about 19% between the two most recent decades, 2000-2009 and 2009-2018. The largest contributor to this increase was a growth in annual CO$_2$-FFI emissions of $6.6\pm0.53$ Gt yr$^{-1}$ decade on decade, complemented by increases of $1.1\pm0.34$

GtCO$_2$eqyr$^{-1}$ in CH$_4$ emissions, $0.42\pm0.30$ Gt yr$^{-1}$ in CO$_2$-LULUCF emissions, $0.25\pm0.15$ GtCO$_2$eqyr$^{-1}$ in N$_2$O emissions, and $0.32\pm0.095$ GtCO$_2$eqyr$^{-1}$ in F-gas emissions. While average annual GHG emissions growth slowed between 2009-2018 compared to 2000-2009 from 2.5% to 1.2%, the absolute increase in average decadal GHG emissions by $8.8\pm0.72$ GtCO$_2$eqyr$^{-1}$ from the 2000s to the last decade (2009-2018) has been the largest since the 1970s – and within all human history as suggested by available long-term data (e.g. Friedlingstein et al., 2020; Hoesly et al., 2018).


More than half of the recent growth in global GHG emissions between 2009 and 2018 came from China (3.2 GtCO$_2$eqyr$^{-1}$) and India (0.95 GtCO$_2$eqyr$^{-1}$) (Figure 8). Among the major emitters, fastest GHG emissions growth was observed for Turkey with average annual rates of 4.2% yr$^{-1}$ between 2009 and 2018, followed by Indonesia (3.8% yr$^{-1}$), Saudi Arabia (3.4% yr$^{-1}$), India (3.2% yr$^{-1}$), Pakistan (3.1% yr$^{-1}$), and China (2.3% yr$^{-1}$). GHG emission reductions achieved by countries over the last

decade are comparatively small even though there is a growing number of countries on sustained emissions reductions trajectories (Lamb et al., 2021b; Le Quéré et al., 2019b). The US showed the largest net anthropogenic GHG emissions reductions of 0.14 GtCO$_2$eqyr$^{-1}$ between 2009 and 2018, resulting from reductions of about the same size in CO$_2$ emissions from a switch from coal to gas in the context of the shale gas expansion. Other countries with decreasing GHG emission levels were Australia (-0.01 GtCO$_2$eqyr$^{-1}$), Germany (-0.02 GtCO$_2$eqyr$^{-1}$), and the United Kingdom (-0.12 GtCO$_2$eqyr$^{-1}$), where the

latter shows the fastest average annual reductions at a rate of 2.9% yr$^{-1}$ in the sample (Figure 8) – in line with some GHG emission reduction scenarios that limit global warming to well below 2°C, but those ones that tend to rely more heavily on carbon dioxide removal technologies (Hilaire et al., 2019; Strefler et al., 2018). Further information on country contributions to GHG emission changes since 1990s – an important reference for UN climate policy – are shown in supplementary Figure SM-3.


Official statistics submitted annually by 43 countries listed in Annex I of the Kyoto Protocol (see Figure 9) to the UNFCCC (hereafter UNFCCC-CRFs) indicate 0.9% lower emissions over the period 1990-2018 (excluding CO$_2$-LULUCF emissions) (UNFCCC (2021), accessed through Gütschow et al.,(2021a)). The vast majority of the Annex I countries, which contributed 33% of the global GHG emissions in 2018 (according to the dataset presented in this paper), report lower total GHG emissions

in 2018 as compared with the data presented here. The total emissions of the Annex I countries in 2018 stand with 17.2 GtCO$_2$eqyr$^{-1}$ according to the national inventories 1.2% lower than the data presented here for the same countries. Both

datasets, however, agree in terms of the average annual growth rates over the last decade (2009-2018), which stood at -0.4% for the total GHG emissions of the Annex I countries. For single countries there is still some divergence in growth rates observed between the national inventories and the dataset presented here (Figure 8, panels b and c). Additional analysis

comparing our data with UNFCCC-CRF inventories for individual (groups of) gases and countries is provided in supplementary Figure SM-3 and Figure SM-4.

Sectoral GHG emissions were either stable or increased between 2009 and 2018. There is high confidence that no substantive GHG emissions reductions were observable for entire sectors at the global level. The most substantial growth was observed in

the metal industry with an average annual growth rate of 3.4% $yr^{-1}$ between 2009 and 2018, followed by the chemical industry (2.5% $yr^{-1}$), road transport (2.0% $yr^{-1}$), electricity and heat (1.9% $yr^{-1}$), and the cement industry (1.7% $yr^{-1}$) (Figure 8, panels d and e). International and domestic aviation, which are small in their contribution to global GHG emissions (and are therefore not shown in Figure 8 e-f), are exhibiting even larger growth rates of 3.8% $yr^{-1}$ (0.69 $GtCO_2eq$ $yr^{-1}$), and 3.7% $yr^{-1}$ (0.39 $GtCO_2eq$ $yr^{-1}$), respectively.

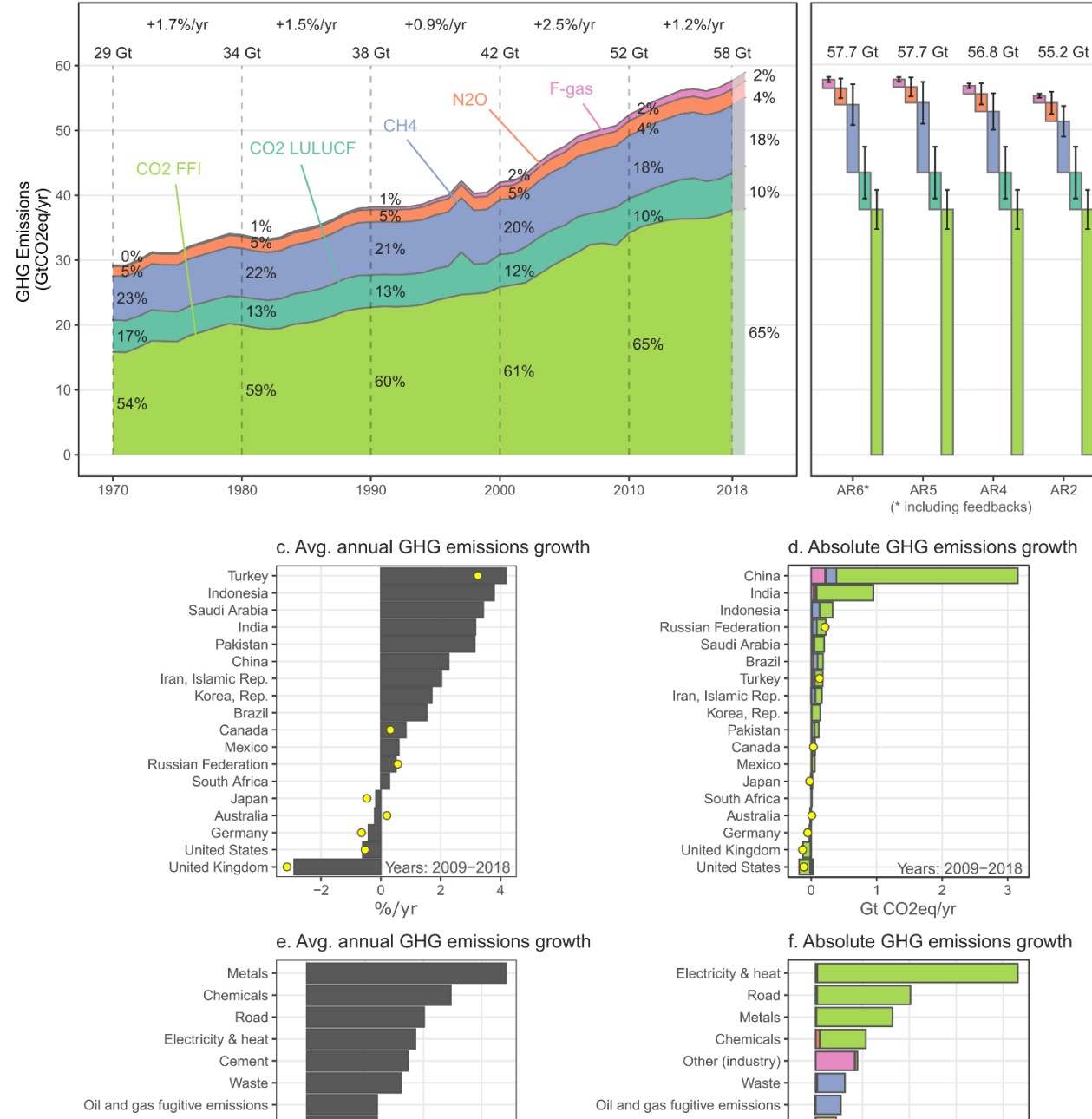

a. Total anthropogenic emissions 1970 - 2019

b. Evolution of GWP100 metric values across assessments

c. Avg. annual GHG emissions growth

d. Absolute GHG emissions growth

e. Avg. annual GHG emissions growth

f. Absolute GHG emissions growth


**Figure 8 - Total anthropogenic GHG emissions (Gt CO₂eq yr⁻¹) 1970-2018 and initial estimates for 2019 as well as country and sector contributions to changes over the last decade (2009-2018)**: CO₂-FFI (light green); CO₂-LULUCF (dark green); CH₄ (blue); N₂O (orange); fluorinated gases (pink); all GHGs (black). Gases are reported in GtCO₂eq converted based on global warming potentials with a 100-year time horizon (GWP-100) from the IPCC AR6 (Forster et al., 2021). Panel a: Aggregate GHG emission trends 1970-2018 with initial 2019 estimate. Average annual growth rates by decade are reported at the top of the figure (in % yr⁻¹). Transparent colour for 2019 estimate indicates its preliminary nature and lower confidence associated with it. Panel b: Waterfall diagrams juxtaposes GHG emissions for 2018 in CO₂eq units using GWP-100 values from the IPCC's AR6, AR5, AR4, and AR2, respectively. Error bars show the associated uncertainties at a 90 % confidence interval (see Section 3). Panels c and d show relative (in % yr⁻¹) and absolute (in GtCO₂eq yr⁻¹) average annual changes in GHG emissions for a selection of the largest emitting countries (contributing 75% of global GHG emissions in 2018) excluding CO₂-LULUCF emissions as uncertainties in our estimates are too high for country-level reporting. The yellow dots represent the emissions data from UNFCCC-CRFs (2021) that were accessed through Gütschow et al. (2021a). Further comparisons with CRF data are provided in Figures SM-3 and SM-4. Panels e and f show relative (in % yr⁻¹) and absolute (in GtCO₂eq yr⁻¹) changes in GHG emissions for a selection of the largest emitting sectors (see Table 2) (contributing 90% of global GHG emissions in 2018).

## 4. 3 Global GHG emissions in 2018

Global net anthropogenic GHG emissions continued to grow and reached $58\pm6.1$ GtCO₂eq in 2018 (Figure 8). In 2018, CO₂ emissions from FFI were $38\pm3.0$ Gt, CO₂ from LULUCF $5.7\pm4.0$ Gt, CH₄ $10\pm3.1$ GtCO₂eq, N₂O $2.5\pm1.5$ GtCO₂eq and F-gases $1.3\pm0.40$ GtCO₂eq. Of the $58\pm6.1$ GtCO₂eq emissions in 2018, 35% (20 GtCO₂eqyr⁻¹) were from energy supply, 24% (14 GtCO₂eqyr⁻¹) from industry, 21% (12 GtCO₂eqyr⁻¹) from AFOLU, 15% (8.6 GtCO₂eqyr⁻¹) from transport, and 5.6% (3.3 GtCO₂eqyr⁻¹) from buildings. In 2018, the largest absolute contributions in GHG emissions were from Asia and Developing Pacific (43%), Developed countries (25%), and Latin America and the Caribbean (10%). China (14 GtCO₂eqyr⁻¹), USA (6.4 GtCO₂eqyr⁻¹), India (3.7 GtCO₂eqyr⁻¹), and the Russian Federation (2.4 GtCO₂eqyr⁻¹) remained the largest country contributors to global GHG emissions, excluding CO₂-LULUCF as we do have not sufficient confidence to report this data at the country level.

In 2018, emissions were $1.1\pm0.11$ GtCO₂eqyr⁻¹ or 1.9% higher than the $57\pm6.0$ GtCO₂eq in 2017. Most of this growth ($0.78\pm0.062$ Gt yr⁻¹, 2.1% yr⁻¹) was related to increases in CO₂-FFI emissions. Also F-gas emissions ($0.067\pm0.020$ GtCO₂eqyr⁻¹, 5.2% yr⁻¹) and CO2-LULUCF emissions ($0.12\pm0.08$ Gt yr⁻¹, 2.1% yr⁻¹) increased significantly, but we assign less confidence in the magnitude of the growth particularly for CO₂-LULUCF due to the high uncertainties attached. Emissions in CH₄ and N₂O were rather stable between 2017 and 2018, with growth rates of 0.8% yr⁻¹ and 0.4% yr⁻¹, respectively. Given prevailing uncertainties there is low confidence that GHG emissions have never been higher than in 2018 as suggested by the data, but high confidence that average annual GHGs emissions have never been higher for a decade than in 2009-2018 (see Friedlingstein et al., 2020; Hoesly et al., 2018).

## 4.4 Fast-track estimates for GHG emissions in 2019

GHG emissions in 2019 are estimated at $59\pm6.6$ GtCO₂eq yr⁻¹. This is 2.2% higher ($1.26\pm0.64$ GtCO₂eq yr⁻¹) than emissions in 2018, and an increase in the annual growth rate compared to 2017-2018 of 1.9% ($1.05\pm0.11$ GtCO₂eq). These estimates are in large parts derived from less complete information and there is less confidence in the exact magnitude. The magnitude of the recent emissions growth is particularly uncertain, because a large portion of emissions growth between 2018 and 2019

(0.91±0.64 Gt yr$^{-1}$, 16.1% yr$^{-1}$) is related to increases in very uncertain $CO_2$-LULUCF emissions estimates. All three bookkeeping models show a consistent trend of increasing emissions in 2019, culminating in an average estimate for net anthropogenic $CO_2$-LULUCF emissions of 6.6±4.6 Gt yr$^{-1}$. This was due to a surge of fire emissions from peat burning, deforestation and degradation, occurring in principally in equatorial Asia and the Amazon, and substantially exceeding average rates in the previous decade (Friedlingstein 2020; GFED4.1s; van der Werf et al., 2017). Anthropogenic fire processes are not captured well by the underlying land-use datasets. SFurther, the 2019 estimate was extrapolated for all three bookkeeping estimates by applying additional information on emissions from equatorial Asia peat fires and tropical deforestation and degradation fires (GFED4.1s; van der Werf et al., 2017) in a similar way (Friedlingstein 2020). This explains the consistent upward trend for all three bookkeeping estimates for 2019.

Non-LULUCF $CO_2$ sources contributed relatively little to the 2019 increase in emissions. $CO_2$-FFI emissions were relatively stable (0.19±0.02 $GtCO_2$eq yr$^{-1}$, 0.5% yr$^{-1}$), as were F-gases (0.4% yr$^{-1}$) while $N_2O$ and $CH_4$ emissions increased with growth rates of 1.3% and 1.1%, respectively. In terms of regions, 89% (1.12 $GtCO_2$eq yr$^{-1}$) of the emissions growth in 2019 occurred in Asia and the developing Pacific, followed by Latin America (0.30 $GtCO_2$eq yr$^{-1}$, 24%) and international shipping and aviation (0.078 $GtCO_2$eq yr$^{-1}$, 6%).

**Discussion**

In this article we provide a comprehensive, detailed dataset for global, regional, national and sectoral GHG emissions from anthropogenic activities covering the last five decades (1970-2019). This is based on the EDGARv6 GHG emissions inventory, but additionally includes a fast-track update to 2019 and data on $CO_2$-LULUCF emissions from three global bookkeeping models. We assess uncertainties in our estimates by combining statistical analysis of the underlying data and expert judgement based on an in-depth review of the literature by each gas. We report uncertainties at a 90% confidence interval ($5^{th}$-$95^{th}$

percentile range). We note that national emissions inventory submissions reported to the UNFCCC are requested to report uncertainty using a 95% ($2\sigma$) confidence interval. The use of this broader uncertainty interval implies, however, a relatively high degree of knowledge about the uncertainty structure of the associated data, which is not present over the emission sectors and species considered here.

Our uncertainty assessment is broadly consistent with previous assessments focussing on all GHGs (Blanco G. et al., 2014; UNEP, 2020), but we provide some important updates. Our evidence-informed uncertainty judgements are higher for $CO_2$-LULUCF ($\pm70\%$ rather than $\pm50\%$) and $CH_4$ ($\pm30\%$ rather than $\pm20\%$) drawing from the global carbon budget (Friedlingstein et al., 2020) and the available literature (Janssens-Maenhout et al., 2019; Solazzo et al., 2021). We note the limited literature on the uncertainties in F-gas emissions in global emissions inventories and recognize the divergence between bottom-up

inventory estimates and top-down atmospheric measurements for individual F-gases. Our revised uncertainty estimate for aggregate F-gas emissions of $\pm30\%$ (rather than $\pm20\%$) reflects the smaller aggregate deviation observed for aggregate F-gas emissions across species. We further acknowledge that we apply the same uncertainty estimates to our fast-track extension to 2019 even though the 2019 estimates themselves will be more uncertain. However, our analysis almost exclusively focusses on the data up to 2018 that is based on full data releases, where our global uncertainty estimates are applied.


     Our analysis involves aggregating GHG emissions into a single unit using GWP-100 values from IPCC AR6, which include climate feedbacks. By doing so we follow the practice taken in UNFCCC inventory reporting and large parts of the literature on climate change mitigation. However, we recognise intense scientific and academic debates about the aggregation of GHGs into a single unit and alternative choices of metrics (Forster et al., 2021) (see Section 3.7). We therefore also use a simple

climate model to assess the warming contribution by the individual groups of gases and find that for the historical period when emissions are growing, the GWP-100 gives a reasonable approximation to the warming contributions, but this is not expected to hold when emissions change trajectory under mitigation. In the absence of a comprehensive uncertainty analysis that covers $CO_2$-LULUCF as well as F-gas emissions, we estimate the overall uncertainty of aggregated GHG emissions by simply adding the individual uncertainties judgements by (groups of) gases in quadrature under the assumption of their independence. Over

time, uncertainties fluctuate between 10% and 14% depending on the composition of gases within the aggregate. Comprehensive uncertainty analysis of EDGAR data covering all GHGs should be performed in the future, building on Solazzo

et al. (2021). We also provide an initial estimate of metric uncertainty arising from the aggregation of individual GHGs into a single unit (see Section 3.7).

We have used the scientific definition for $CO_2$-LULUCF emissions, in line with our intention to identify GHG fluxes attributable to human activities and in line with the GCP's (Friedlingstein et al., 2020) and IPCC WG1's (Canadell et al., 2021) approach that split natural from anthropogenic drivers. We have acknowledged that this differs from national GHGI (Grassi et al., 2018) or inventory data provided by FAO (Tubiello et al., 2021), which should be used if consistency in definition with, e.g., the UNFCCC inventories is required. Net $CO_2$-LULUCF emissions estimates are substantially smaller based on inventory

data over managed land, because the environmental drivers (e.g. $CO_2$ fertilisation) of terrestrial sink on managed land are attributed to anthropogenic emissions in UNFCCC inventories. This highlights the potential of land in emissions reduction efforts: on the one hand, net emissions from land-use activities should be minimized by reducing gross emissions (e.g., by stopping deforestation and degradation) and increasing gross removals (e.g., by reforestation) (Roe et al., 2019); on the other hand, vegetation acting as a natural sink to anthropogenic $CO_2$ emissions should be retained, be it via managed land, as in the

inventories, or via pristine vegetated lands.

While the distinction between the scientific approach (drivers) using bookkeeping models and the inventory approach (areas) is clear, and methods to map between approaches have been suggested (Grassi et al., 2018, 2021), the attribution of environmental and anthropogenic changes differs between methods. Further, it should also be mentioned that system

boundaries partly differ across datasets, and FAOSTAT data (Conchedda and Tubiello, 2020) is currently limited to $CO_2$ fluxes related to forests and emissions from drainage of organic soils under cropland/grassland, excluding other managed land or agricultural conversions. In principle, bookkeeping and DGV models include all fluxes, but are often coarse in their description of management, which observation-based approaches capture (Arneth et al., 2017). Several authors (Grassi et al., 2018; Obermeier et al., 2020; Pongratz et al., 2014) have shown the strong dependence of $CO_2$-LULUCF emissions estimates on the

time a certain land-use change event happened to occur, if environmental changes are represented transiently over time, as is the case for typical simulations with dynamic global vegetation models. This dependence is eliminated by using bookkeeping estimates, as done here.

Comparisons with other global emissions inventories highlights the comprehensive nature of our dataset covering

anthropogenic sources of GHG emissions. However, there are still some important data issues. In particular, F-gas emissions estimates for some individual species in EDGARv5 and v6 do not align well with atmospheric measurements and the F-gas aggregate emissions over the last decade either overestimate top-down estimates by around 30% (EDGAR v5) or under-estimate them by around 10% (EDGAR v6). Furthermore, EDGAR and official national emission reports under the UNFCCC do not comprehensively cover all relevant F-gases species. In particular, chlorofluorocarbons (CFCs) and

hydrochlorofluorocarbons (HCFCs), which are regulated under the Montreal Protocol, have historically contributed more to

CO$_2$eq emissions as well as observed warming than the F-gases included in our study. There is an urgent need to dedicate more resources and attention to the independent improvement of F-gas emission statistics, recognizing these current shortcomings and their important role as a driver of future warming. We also find a need for more transparent methodological documentation of some of the available inventories – particularly for F-gas emissions. Moreover, recent work on the global methane budget (Saunois et al., 2020b) and the global nitrous oxide budget (Tian et al., 2020) further suggest discussions on whether global inventories should be further expanded in terms of their reporting scope.

Our analysis of global, anthropogenic GHG emission trends over the past five decades (1970-2019) highlights a pattern of varied, but sustained emissions growth. There is high confidence that global anthropogenic GHG emissions have increased every decade. Emission growth has been varied, but persistent across different (groups of) gases. While CO$_2$ has accounted for almost 75% of the emission growth since 1970 in terms of CO$_2$eq as reported here, the combined F-gases have grown much faster than other GHGs, albeit starting from very low levels. Today, they make a non-negligible contribution to global warming (see Figure 4). However, our results are focussed on F-gases (HFCs, PFCs, SF$_6$, NF$_3$) that are regulated under the Paris Agreement. Other species such as CFCs and HCFCs that are regulated under the Montreal Protocol had much larger cumulative warming impacts over time (see Figure 4), but are not considered here as is common in GHG emissions inventory discussions. A fuller consideration of all F-gas emissions together, independent of the regulatory framework, would change both magnitude and their development over time. Overall, aggregate CO$_2$eq emissions from F-gases would more than double in 2018, but emissions would be largely decreasing over time due to large and steady cumulative emissions reductions in species regulated under the Montreal Protocol.

There is high confidence that global anthropogenic GHG emissions levels were higher in 2009-2018 than in any previous decade and GHG emissions levels have grown across the most recent decade. While average annual GHG emissions growth slowed between 2009-2018 compared to 2000-2009, the absolute increase in average decadal GHG emissions from the 2000s to the 2010s has been the largest since at least the 1970s when the dataset starts – and within all human history as suggested by available long-term data (e.g. Friedlingstein et al., 2020; Gütschow et al., 2016, 2021b; Hoesly et al., 2018). While there is a growing number of countries today on a sustained emission reduction trajectory (Lamb et al., 2021b; Le Quéré et al., 2019a), GHG emissions are growing over time for all global sectors and sub-sectors in our dataset mirroring global GHG emissions trends that are characterized by distinct patterns of development and industrialization. It is therefore important to study the drivers of these reductions as well as patterns of emission growth in more detail at the regional level (Lamb et al., 2021a) and systematically evaluate the impact of climate-relevant policies on regional drivers and trends.

There is a growing availability of global datasets on anthropogenic emissions sources over the last 10-20 years (see Table 1). However, such global emission inventories often heavily rely on relatively simple Tier-1 estimation methods and few use more complex Tier-2 or Tier-3 methods. Comparison of our estimates with UNFCCC-CRFs by Annex I countries shows

considerable discrepancies for some gases and countries (see Figure 8, Figure SM-3, Figure SM-4). On aggregate, there is a clear trend towards smaller values for GHG emission reductions and larger values for GHG emission increases in our dataset. Further work needs to be done to fully appreciate underlying differences, as has been done, for example, for $CO_2$ emissions (Andrew, 2020a) and for Europe across all GHGs (Petrescu et al., 2020b, 2021b, 2021a). Figure 9 further highlights the lack of recent official national emissions inventories for many non-Annex I countries. The BURs are also associated with less

stringent reporting requirements in terms of sector, gas and time coverage (Deng et al., 2021; Gütschow et al., 2016). This highlights the important role of global inventories such as EDGAR, CEDS, PRIMAP-hist, FAOSTAT or those from IEA or BP among others  that are equally as comprehensive in scope to those from Annex I countries. Despite the importance of high-quality emission statistics for climate change research and tracking progress in climate policy, our analysis here emphasises considerable prevailing uncertainties and the need for improvement in emission reporting. Additionally, there are significantly

fewer independent estimates for full GHG accounting, in contrast to fossil $CO_2$ emissions. In sectors where production efficiencies are changing rapidly, as is often the case in developing countries, using emission estimates based on Tier-1 methodologies may mischaracterise trends as both activity data and emission factors change over time (Wilkes et al., 2017).

Moving confidently towards net-zero emissions requires high quality emissions statistics for tracking countries' progress based

at least on Tier-2, if not on complex Tier-3 estimation models using comprehensive, country-specific activity data and emissions factors or atmospheric inversions (IPCC, 2019). This would also support the formulation of more nuanced climate policy goals that reflect changes in emissions intensities as entry points for more comprehensive and ambitious targets to reduce absolute emissions. However, underpinning such approaches with robust evidence requires the collection of a range of country-specific activity data and development of adequate statistical infrastructure for all countries of the world (FAO and

GRA, 2020). In parallel, it might be a pragmatic way forward to continue and intensify work on comprehensive, up-to-date global emissions inventories such as EDGAR or CEDS as well as synthetic datasets as presented here or PRIMAP-hist. Future extensions of this work could update country and sector-specific data from UNFCCC inventories wherever possible and available. It could also make sense to add missing emissions components – particularly, in non-$CO_2$ emissions from AFOLU – and develop fast-track methods to extend the inventories from the last available inventory year to the most recent year.

Keeping global warming well below 2°C and pursuing efforts towards 1.5°C requires dedication and cooperation between countries: working together on a robust evidence base in GHG emissions reporting provides one important and often underappreciated step.

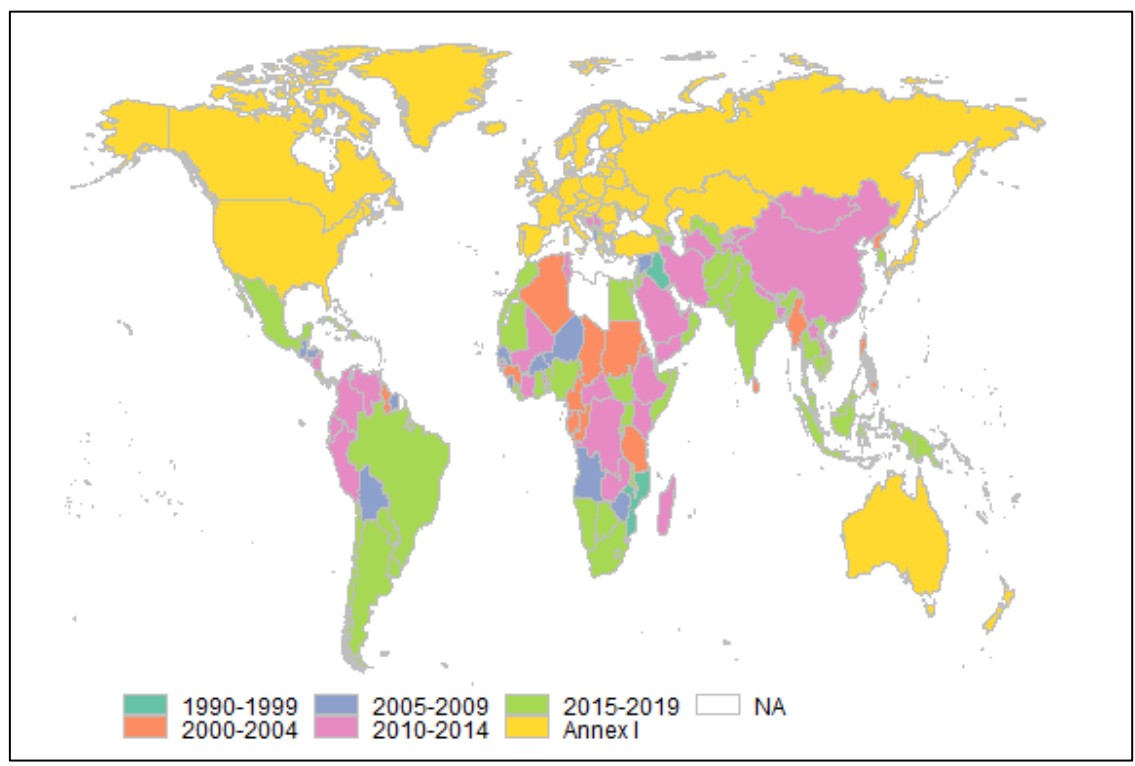


**Figure 9 - Overview of most recent GHG emission inventories submitted to the UNFCCC:** The map captures the last year for which emission inventories were conducted and published by the UNFCCC on their website (as of 28 September 2021) including CRFs, BURs, and NCs. Annex I countries, according to the UNFCCC definition, have reported their last inventories for 2019 (UNFCCC, 2021). Updated from Janssens-Maenhout et al. (2019)


**Data availability**

The emissions dataset used for this study (Minx et al. 2021) is available at https://doi.org/10.5281/zenodo.5053056

[NOTE TO REVIEWERS: Data on $CO_2$ emissions from fossil fuel combustion and industry, methane emissions and nitrous
oxide emissions are from the most recent EDGARv6 data. As EDGARv6 data is still being compiled for F-gases, this
manuscript contains EDGARv5 estimates for these, but we will update to EDGARv6 during the revision process. This
procedures has been agreed upon with David Carlson – one of the chief editors of the journal – before manuscript submission]

**Author contributions**

JCM and WFL designed the research. WL, ND, RMA, GPP, MR and PMF generated the figures with support by all other
authors (JCM, JGC, MC, DG, JO, JP, AR, MS, SJS, ES, HT). WFL, ND, RMA, GPP, MR and PMF carried out the required
computations. JCM led on the analysis in collaboration with all authors (WFL, RMA, JGC, MC, ND, PMF, DG, JO, GPP,
JP, AR, MR, MS, SJS, ES, HT) JCM led on the writing of the manuscript in collaboration with all authors (WFL, RMA,
JGC, MC, ND, PMF, DG, JO, GPP, JP, AR, MR, MS, SJS, ES, HT).

**Funding acknowledgements**

The authors would like to thank Yang Ou for helpful comments on the manuscript and Eduardo Posada as well as Lucy Banisch
for their help with compiling the information for Table 1 and Figure 9, respectively. JCM and WL acknowledge funding from
the German Ministry for Education and Research (grant no. 01LG1910A). RMA and GPP acknowledge funding from the
European Commission Horizon 2020 projects VERIFY (grant no. 776810) and CoCO2 (grant no. 958927). PMF acknowledges
funding by the European Union's Horizon 2020 Research and Innovation Programme under grant no. 820829 (CONSTRAIN).
HT acknowledges funding support from the National Science Foundation (Grant #: 1903722). JGC acknowledges the support
of the Australian National Science Program – Climate Systems Hub. MR is supported by UK Natural Environment Research
Council grants NE/N016548/1, NE/M014851/1 and NE/I021365/1.

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
