# Peer review of "A comprehensive and synthetic dataset for global, regional and national greenhouse gas emissions by sector 1970-2018 with an extension to 2019"

_Earth System Science Data, 2021_

## Author Response (AR1)

**General responses**

First of all we would like to thank all three reviewers for their very constructive and helpful comments, which have helped to further improve the quality, transparency and consistency of the document. Before we respond to their specific comments, we would like to note some general changes that we have undertaken since the first version of the paper.

- We have updated all Tables and Figures based on a new, updated version of the data:
    - EDGAR data is now consistently from EDGARv6 for the time period 1970-2018 – also for fluorinated gases. In EDGARv6 $NF_3$ is added as another F-gas species that is regulated under the Paris Agreement.
    - We use the upcoming EDGARv6 fast-track for extending $CO_2$-FFI emissions to 2019. $CO_2$-LULUCF emissions are full release estimates from the Global Carbon Budget 2020 (Friedlingstein et al., 2020). Non-$CO_2$ emissions are extended based on Olivier and Peters (2020) fast-track as provided by Olivier and Peters (2020). Procedures are described in the methodology.
- We improved the consistency of data comparison Tables and Figures:
    - In Figure 1 we use the different data sources more consistently across the different gases. For example, PRMIAP-hist estimates are now also provided for $CH_4$ and $N_2O$. FAOSTAT data is used for both CH4 and N2O etc or EPA data is used for all non-$CO_2$ GHGs.
    - We added a comparison of different EDGAR versions also for fluorinated gases as new Figure SM.2. We include also the EPA data in there (as we do not think that it is worthwhile to add a f-gas panel to Figure 1 due to limited data availability).
- We restructured the result discussion to better reflect the data quality of our dataset. We focus on the time period 1970-2018, where we have a full EDGARv6 data release and for which our uncertainty judgements apply:
    - 4.1 - Global anthropogenic GHG emissions for 1970-2018
    - 4.2 - Global anthropogenic GHG emissions for the last decade 2009-2018
    - 4.3 - Global anthropogenic GHG emissions in 2018

    In Figures of the result section that still contain the 2019 estimates, we add a transparency layer to flag that this data is different from the rest.

    We add a new section 4.4 that provides fast-track estimates for 2019.

    - 4.4 - Global anthropogenic GHG emissions in 2019: initial fast-track estimates

    We highlight in the discussion that these estimates are more uncertain – even though we still apply the same global uncertainty estimates.

- Due to the new F-gas data, we edited section 3.5 more heavily.
- We tried to improve clarity and language throughout the manuscript.

**Reviewer 1**

As described in the title, the authors made a comprehensive overview and summary of the various global estimates of GHGs by country and sectors over the last five decades. This synthesis is based mainly on estimates from the EDGAR v6.0 (and v5.0) data, GAINS model, and three bookkeeping models for the CO2 LULUCF. It focuses only on the anthropogenic emissions of CO2 (FF and LULUCF), CH4, N2O and some fluorinated gases. Some top-down HFCs estimates from WMO are compared to bottom-up estimates. For global GHG emissions the authors compared their values to the UNFCCC (2021) estimates (mainly for Annex-I countries). A general comment on this: Is the data you define as CRFs the one officially submitted by countries to UNFCCC? I don't see it referenced, but instead I see PRIMAP, perhaps authors could explain better. If you used both add please a reference to UNFCCC (2021), with link to the data.

Yes. We refer to the official CRF submissions by countries. However, we indeed access this through PRIMAP-CRF for ease-of-use and made sure that this data is fully consistent. We make this clear now by referencing the data in the following whenever it is used (main manuscript and supplementary material): "(UNFCCC (2021), accessed through Gütschow et al. (2021))"

Of high interest is the calculation of uncertainties pertaining to different gases and sectors, as well as global GWPs in CO₂eq. The authors used the 90% confidence interval. I would be interested to know, if authors compare with UNFCCC, why they did not use the 95% confidence interval as done in reporting?

We provide comparisons of our emission estimates with UNFCCC-CRF data in Figures 8, Figure SM.2 and Figure SM.3. But we do not compare uncertainty assessments provided by CRFs. The assessment provided here is restricted to global estimates, but considers multiple lines of evidence in the tradition, for example, of Friedlingstein et al. (2020), Saunois et al. (2020) or Tian et al. (2020). As such the error propagation methodologies as applied in the context of UNFCCC reporting or for EDGAR by Solazzo et al. (2020) or Janssens-Maenhout et al. (2019) is only one line of evidence that informs our expert judgement.

We report at the 90% confidence interval, because we are very wary that a 2σ reporting would imply an even higher degree of knowledge about the uncertainty structure of the associated data particularly regarding the distribution of uncertainty in the tails of the probability distributions. Such a high degree of knowledge is not present over all the emission sectors and species considered here. As such we settle cautiously between the 1σ estimate commonly used by the Global Carbon Budget with the broader remit of its assessment (i.e. the budget approach taken) and the emissions inventory community with its particular approach to uncertainty assessments. We therefore considerably extended the explanation of our approach in the introduction to the uncertainties section (3) as follows:

"This sections provides an assessment of uncertainties in greenhouse gas emissions data at the global level. The uncertainties reported here combine statistical analysis, comparisons of global emissions inventories and expert judgement of the likelihood of results lying outside a defined confidence interval, rooted in an understanding gained from the relevant literature. At times, we also use a qualitative assessment of confidence levels to characterize the annual estimates from each term based on the type, amount, quality, and consistency of the evidence as defined by the IPCC (IPCC, 2014).

Such a comprehensive uncertainty assessment covering all major groups of greenhouse gases and considering multiple lines of evidence has been missing in the literature. The absence has provided a serious challenge for a transparent, scientific reporting of GHG emissions in climate change assessments like those by IPCC's Working Group III or the UN Emissions Gap Report that have only more recently started to even deal with the issue (e.g. Blanco G. et al., 2014; UNEP, 2019). Most of the available studies in the peer-reviewed literature using multiple lines of evidence for their assessment have focused on individual gases like in the Global Carbon Budget (Friedlingstein et al., 2020), the Global Methane Budget (Saunois et al., 2020) or the Global Nitrous Oxide Budget (Tian et al., 2020) or covered multiple gases, but mainly considered individual lines of evidence (Janssens-Maenhout et al., 2019; Solazzo et al., 2021).

We adopt a 90% confidence interval (5th-95th percentile) to report the uncertainties in our GHG emissions estimates, i.e., there is a 90 % likelihood that the true value will be within the provided range if the errors have a Gaussian distribution, and no bias is assumed. This is in line with previous reporting in IPCC AR5 (Blanco et al., 2014; Ciais et al., 2014). We note that national emissions inventory submissions reported to the UNFCCC are requested to report uncertainty using a 95% ($2\sigma$) confidence interval. The use of this broader uncertainty interval implies, however, a relatively high degree of knowledge about the uncertainty structure of the associated data, particularly regarding the distribution of uncertainty in the tails of the probability distributions. Such a high degree of knowledge is not present over all the emission sectors and species considered here. Note that in some cases below we convert $1\sigma$ uncertainty results from the literature to a 90% confidence interval by implicitly assuming a normal distribution. While we do this as a necessary assumption to obtain a consistent estimate across all GHGs, we note that this itself is an assumption that may not be valid. We have made use of the best available information in the literature, but note that much more work on uncertainty quantification remains to be done. Using IPPC uncertainty language, we cannot place high confidence in the robustness of most existing uncertainty estimates."

Also linked to this topic, you use only the methodology of Solazzo et al., 2021 or the exact values for EDGAR uncertainties as calculated by Solazzo et al 2021? As far as I know Solazzo et al. (2021) calculated uncertainties for EDGAR only for 2015 using the 95 % confidence of a log-normal distribution; it is not clear if in this study you used those numbers for EDGAR uncertainties because a) you state that you use the 90% method b) you study 1970-2019.

We consider Solazzo et al. (2021) as one important input for our broader uncertainty assessment using multiple lines of evidence. Note that Solazzo et al. (2021) is based on EDGARv5 and covers only the time period up to 2015. It also does not cover F-gas emissions. Finally, we discuss the particular approach undertaken by Solazzo et al. (2021), its underlying assumptions and implications for resulting estimates in the introductory section to the uncertainty section (3). We tried to clarify the approach taken. Please refer to the text provided in the answer to the previous comment.

Overall the paper is well written and has a clear structure and the authors did a good job in identifying and explaining the discrepancies between datasets. The authors provide also some explanations regarding the rates of change between different periods and confirm sustained emissions growth but they recommend more studies on drivers, and I totally agree with them. An interesting conclusion is that are no global sectors that show sustained reductions in GHG emissions. I think this is normal given the large differences in terms of industrialization and development between countries. This should be investigated more locally (region, continent)

where local policies could affect the emission levels e.g. highlighted in the last two European EU27+UK syntheses by Petrescu et al 2021a,b.

Thank you so much for all your efforts and helpful comments. On the last point: we flag this up in the discussion and highlight the importance of more local investigations of relevant drivers at more regional scales. Lamb et al. (2021a) already provides a nice starting point to build upon. The relevant text in the discussion reads:

"While there is a growing number of countries today on a sustained emission reduction trajectory (Lamb et al., 2021b; Le Quéré et al., 2019a), GHG emissions are growing over time for all global sectors and sub-sectors in our dataset mirroring global GHG emissions trends that are characterized by distinct patterns of development and industrialization. It is therefore important to study the drivers of these reductions as well as patterns of emission growth in more detail at the regional level (Lamb et al., 2021a) and systematically evaluate the impact climate-relevant policies on regional drivers and trends."

I recommend it for publication, subject to some minor changes and suggestions, as highlighted below.

**Specific comments:**

L28-29: perhaps state that you use both bottom-up and top-down estimates (at least for HFCs)

Thank you. We made highlighted this in the abstract now. The relevant sentence reads: "We assess the uncertainties for global greenhouse gas at the 90% confidence interval (5th-95th percentile) by combining statistical analysis and comparisons of global emissions inventories and top-down atmospheric measurements with an expert judgement informed by the relevant scientific literature."

L39: add "Global" GHG emissions

Accepted and added.

L40: add CO2 after Gt

Accepted and added

L49: could you add a value in brackets with how much emissions slowed down between 2010-2019 compared to 2000-2009?

Accepted. The sentence now reads: "While average annual GHG emissions growth slowed between 2009-2018 (1.2% $yr^{-1}$) compared to 2000-2009 (2.5% $yr^{-1}$), the absolute increase in average decadal GHG emissions from the 2000s to the 2010s was the largest since the 1970s, and within all human history as suggested by available long-term data."

L51: higher rates like how much between 2018 and 2019?

We removed this part of the abstract as it was rather lengthy.

L55: you state that "there is (worldwide?) a growing number of countries today on a sustained emission reduction trajectory" do you exemplify this somewhere in a table or have reference for it? Would be interesting to see how much of these countries are non-Annex I

> There is dedicated analysis on this in the literature and we cite some key pieces (Lamb et al., 2021b; Le Quéré et al., 2019b). However, we rephrased the sentence in a way that it is supported by the evidence presented in this manuscript (e.g. Figure 8). The sentence now reads: "There is a number of countries that have reduced GHG emissions over the past decade, but these reductions are comparatively modest and outgrown by much larger emissions growth in some developing countries such as China, India and Indonesia."

L84: I agree that non-Annex I countries lack well developed statistical infrastructure but they do submit biannual reports, see  Z Deng et al 2021 where they use global BURs UNFCCC global data. https://essd.copernicus.org/preprints/essd-2021-235/

> We agree that the statement was not well balanced. We added the Deng et al. (2021) as well as the Gütschow et al. (2016) reference and rephrased the text in the following way:

> "GHG emissions reporting under the United Nations Framework Convention on Climate Change (UNFCCC) provides reliable, comprehensive and up-to-date statistics for Annex I countries across all major GHGs and sectors. Non-Annex I countries – except least developed countries and small island state for which this is not mandatory - provide GHG emissions inventory information through biennial update report, but with much less stringent reporting requirements in terms of sector, gas and time coverage (Deng et al., 2021; Gütschow et al., 2016). As a result, many still lack a well-developed statistical infrastructure to provide detailed reports (Janssens-Maenhout et al., 2019)."

L88: EDGAR add versions

> We made a clear reference to EDGARv6 as the core building block here. We add more details in the subsequent methodology section.

Table 1: Time period for EDGAR v6.0: check the year "1970-2015 for other GHG" I think is 2018 see also your statement on L122

> Thanks for this. We updated the table.

Table 1 and all the text: is Global Carbon Budget GCB or Global Carbon Project GCP, be consistent

> In the table the short name was corrected to GCB. In the text we use both, GCB and GCP to distinguish between the emissions dataset and its producing institution.

Table 1: for GAINS you could add some recent refs as well (Höglund-Isaksson (2012), Höglund-Isaksson (2017), Höglund-Isaksson et al. (2020), Gomez-Sanabria et al. (2018), Winiwarter et al. (2018) and some contact info for the last column

> We added additional references. Thanks

Table 1: FAOSTAT inventory has been updated to 2019 this year, you have 1990-2017 why?

*We updated the FAOSTAT data as well. This is now also visible in Figure 1. Thanks.*

Table 1: GFED emissions should be reference everywhere in the text as van der Werf et al., 2017 because Giglio is only providing the burned area (AD), but van der Werf is providing emissions.

*Thanks. We changed the reference…*

and please add in the last column the link: https://www.geo.vu.nl/~gwerf/GFED/GFED4/

*… and added the suggested link. Thanks!*

L101: add Table for sectors disaggregation

*We have such a Table already. See Table 2 of the manuscript.*

L129: I would replace "as is common" with 'as commonly used in IPCC reports"

*Accepted. We have changed the language accordingly.*

L130: IPCC here refers to AR5 but now there is AR6 available, could you perhaps update to it?

*Rejected. We are citing the AR5 IPCC Working Group III on Climate Change Mitigation report where emissions inventory data is discussed. The AR6 report will only be released after the approval plenary at the end of March 2022 and, therefore, is not available for citation in this paper.*

L220: Petrescu et al references are not updated, should be 2020, 2021a and 2021b

*Accepted. We changed the references accordingly.*

L233: same here, should be Petrescu et al 2021a,b

*Accepted. We changed the references accordingly.*

L235: regarding Solazzo et al. (2021) see my question in the general comments paragraph

*As highlighted in the response to your general comment, we have further clarified and motivated our approach taken for the uncertainty assessment.*

L245: you refer here for CH4 right? Please add CH4

*No. This is a generic statement regarding approaches to the assessment of uncertainties. It is not specific to methane emissions.*

L252, 297, 324, 533, 610, 924: I am wondering why you reference everywhere in the text Blanco G. et al instead of Blanco et al?

*Thanks. This was a problem with the reference. Resolved now.*

L259-L266: this paragraph has a different line spacing

Resolved. Thanks.

L286: can you give few examples for some countries?

Accepted. We added three examples and provide now an explicit reference to Figure SM-3, which contains this information.

Figure 1 the resolution is a bit low, not so clear to read the text and see the lines

Accepted and done.

L340: Caption Figure 1: Is Tubiello 2013 last reference for latest FAOSTAT N2O inventory? Was updated recently

Accepted. Added the more recent references.

Table 4: H&N *Sub-grid scale*: the no with down arrow means that is not represented in the model and therefore it decreases emissions? Why *Pasture conversion* has 'no, yes' next to arrows?

Accepted. We clarified this now in the sub-heading of the table. It reads: "Representation of processes (Arrows: indicative effect on AFOLU CO2-LULUCF emissions)"

"*Distinction rangeland vs Pasture*": how are these two different? Can you reference the definitions you used for rangeland and pasture?

The LUH2 land-use forcing underlying e.g. BLUE and OSCAR separate out rangelands from the more general pastures/managed grassland category. This is done based on an aridity index and population density as detailed in Klein Goldewijk et al. (2017; https://doi.org/10.5194/essd-9-927-2017). While pastures imply land use as a grassland, rangelands often do not change the natural vegetation to grassland (e.g., shrubland browsing), but may nevertheless degrade the existing vegetation carbon stocks. To constrain the models' interpretation of whether rangeland implies the original natural vegetation to be transformed to grassland or not, a forest mask was provided with LUH2 for the 2020 GCP budget; forest is assumed to be transformed to grasslands, while other natural vegetation remains (in case of secondary vegetation) or is degraded from primary to secondary vegetation (Ma et al., 2020). This is implemented in BLUE and described in Friedlingstein et al. (2020). We point the reader to this reference now in a footnote in the table, which reads:

"based on rangeland-pasture distinction of the HYDE dataset (Klein Goldewijk et al., 2017) and forest cover map of Hurtt et al. (2020); see Friedlingstein et al. (2020) for details"

L424, 503: replace Giglio with van der Werf et al 2017

Accepted. Done. Thanks.

L425 and all text: check the 2 from $CO_2$ to be subscript

Amended, and entire text checked

L426: to your statement "inventories of anthropogenic emissions are not completely independent" you could cite Petrescu et al 2020 AFOLU paper Figure 4

Good idea – added!

L445 and in all text: the unit Mt per year should be Mt $yr^{-1}$

Amended (this was the only occurrence in the text)

Table 5: next to Saunois et al 2020 uncertainty for TD estimates I think Bergamaschi et al 2018 has an uncertainty estimate as well https://op.europa.eu/en/publication-detail/-/publication/4aff4499-8322-11e8-ac6a-01aa75ed71a1/language-en

True. There are other studies providing uncertainty estimates of different kinds. We have clarified the intention of the table further in the caption. We added: Note that this tables provides uncertainty estimates from some of the key literature based on different methodological approaches. It is not intended to be an exhaustive treatment of the literature.

Table 5: please state how EDGAR uncertainties were calculated (see d and e notes, are from two sources)

References to the underlying papers are provided.

L476: v4.3.2

Accepted. Done.

L490: I would say: "The emissions from the agriculture sector have four components ..."

Sentence amended

L491: how are ocean waters anthropogenic sources? You refer here to the national ocean/sea waters belonging to the countries beyond coasts or ?

The only anthropogenic emission from open oceans is indirect emission - Atmospheric nitrogen deposition on oceans, which is accounted in Tian et al. (2020).

L501: the better reference for N2O from GAINS is "Winiwarter, W., Höglund-Isaksson, L., Klimont, Z., Schöpp, W., and Amann, M.: Technical opportunities to reduce global anthropogenic emissions of nitrous oxide, Environ. Res. Lett., 13, 014011, https://doi.org/10.1088/1748-9326/aa9ec9, 2018."

Accepted. We replaced the reference.

L502: FAOSTAT-N2O

Accepted. Done

L530: would be nice to see values in brackets for this comparisons between sector uncertainties

Accepted. Done.

Figure 3 and in all text: consistency between using v5.0 v6.0 and v5 and v6 for EDGAR releases

We ensured consistent representation of version numbers throughout the document.

Figure 3 caption: add top-down (WMO)

Accepted. Done.

Figure 3 caption and all text: sometimes you use words for sigma sometimes symbol, constancy

Captions of Figure 2 and 3 amended. No other occurrences of the word found

L576: which EDGAR v4? 4.3.2? please add reference

This is correct. It refers to EDGARv4. See Rigby et al. (2010). As there are many different EDGARv4, we specify in this case 4.0.

L581: "EDGAR had previously" which study? Reference please

We reworked this paragraph with clear referencing. The relevant sentence now reads: "For some PFCs (e.g., CF4, C2F6), it was previously noted that some assumptions within EDGAR had been validated against atmospheric observations , hence EDGAR might be considered a hybrid of top-down and bottom-up methodologies for these species (Mühle et al., 2010)."

L588: I would add after (Figure 3), we note that in the left panel

amended

L592: after 90% I would add (right panel)

Here also the left panel of the figure is relevant. We added a reference to it. Thanks.

L597 and all text: equivalents sometimes is in words, sometimes eq, consistency

Amended in all text, upon first use abbreviation added in parentheses

L608: "we are taking"

Thanks. Amended.

L617: sometimes GWP-100 sometimes GWP 100

Thanks. Dash added to all occurrences, where it was missing

L623, 647, 946 and all text: please use or methane or CH4

> Thanks. Consistently amended to $CH_4$.

L623: add reference for the GWP values of 86t CO2eq

> We added the reference to WGI AR6 chapter 7 (Forster et al., 2021) at the end of the sentence.

L633, 634 and all text: sometimes you write Second, Fifth Assessment Report or only AR2 or AR4 or AR5, consistency. Also, are you thinking during the review process to update to AR6?

> Consistently amended to AR2, AR4 and AR5

> Yes we updated to AR6 GWP-100 values from Forster et al. (2021).

Figure 4 caption: 90 % confidence interval (or 5%-95% percentile range)

> Thanks. Amended

L706: "greenhouse gas" please check all text and write GHG or in words

> Consistently amended to GHG.

L718: AR5

> Amended. Thanks.

Table 7: CO2 FFI column Growth 1.0%: I understand this is the growth between 2010 and 2019 (so in 2019 we see 1% growth compared to 2010?) or is the average growth (2010-2019) with respect to other average growth from a previous period??

> We amended the table header, repeating "average annual emissions growth" to make this clearer.

L724, 742, 837, 892, 945, 957, 964: GHGs, GHG etc.

> Amended. Thanks.

L737: H&N (FAOSTAT)?

> We acknowledge that this information is provided via FAOSTAT. However, we use FAOSTAT here for the reference to the emissions data, not for the land use area, while the latter is called FAO/FRA in Table 4. We would therefore like to preserve the way of referencing. No change.

L798: CMIP6 appears here for the first time in the text, can you please add a link or reference?

The sentence contains already multiple references, but we added the overview paper by Eyring et al. (2016).

L857: please add UNFCCC (2021) and reference.

Reference added. We further make clear that we access the data through Gütschow et al. (2021). Consistency of the data checked.

L863: "inventories *in 2021* was on..."

With the update of the data, the sentence changed in a way, that the suggestion could no longer be adopted.

Caption Figure 8 is bold

Changed to normal font. Thanks.

L921: well characterized? Or better say correlated?

Based on the high-level comments with regard to the treatment and discussion of uncertainties, we have revised the language here. The relevant section now reads:

"We report uncertainties at a 90% confidence interval (5th-95th percentile range). We note that national emissions inventory submissions reported to the UNFCCC are requested to report uncertainty using a 95% (2σ) confidence interval. The use of this broader uncertainty interval implies, however, a relatively high degree of knowledge about the uncertainty structure of the associated data, which is not present over the emission sectors and species considered here."

L946: comprehensive *global* assessments

Amended. Thanks.

L947: N2O

We are not sure what to do here. No change.

L962: CFCs and HCFCs: are there any studies where these two missing gases from your study are given a % contribution to global warming?

We provide this information in Figure 4 and provide a cross-reference now. We have revised the entire passage for clarity. It now reads: "While $CO_2$ has accounted for almost 75% of the emission growth since 1970 in terms of $CO_2$eq as reported here, the combined F-gases have grown much faster than other GHGs, albeit starting from very low levels. Today, they make a non-negligible contribution to global warming (see Figure 4). However, our results are focussed on F-gases (HFCs, PFCs, SF6, NF3), which are regulated under the Paris Agreement. Other important species such as CFCs and HCFCs regulated under the Montreal Protocol had much larger cumulative warming impacts over time (see Figure 4), but are not considered here as common in GHG emissions inventory discussions. A full consideration of all F-gas emissions including those species would change both magnitude and their development over time. Overall, $CO_2$eq emissions in 2018 would more than double, but

emissions would be largely decreasing over time due to large and steady cumulative emissions reductions in species regulated under the Montreal Protocol."

L969: can you give some hypothetic reasons for this high uncertain increase? References? Values for uncertainty?

Our language was confusing. There is not an increase in uncertainty, but most of the most recent GHG emissions increases come from highly uncertain $CO_2$-LULUCF emissions. We corrected that.

L978: You mention here Tier 2 and 3 UNFCCC CRF data, if you are using the numbers from the CRFs (emissions) they should be almost all calculated according Tier-1 definitions EM=EFxAD with IPCC default EFs, higher Tiers are used only by few countries as well as for reporting uncertainties (see NIRs and their Annexes) where few countries report Tier 2 (Monte Carlo). I think here you could present a table (in SM) where you specify which countries use which Tiers

We rather opted for a more careful language in this part of the text. It now reads:

"There is a growing availability of global datasets on anthropogenic emissions sources over the last 10-20 years (see Table 1). However, such global emission inventories often heavily rely on relatively simple Tier-1 estimation methods and few use more complex Tier-2 or Tier-3 methods. Comparison of our estimates with UNFCCC-CRFs by Annex I countries shows considerable discrepancies for some gases and countries (see Figure 8, Figure SM-X). On aggregate, there is a clear trend towards smaller values for GHG emission reductions and larger values for GHG emission increases in our dataset. Further work needs to be done to fully appreciate underlying differences, as has been done, for example, for CO2 emissions (Andrew, 2020a) and for Europe across all GHGs (Petrescu et al., 2020b, 2021b, 2021a). Figure 9 further highlights the lack of recent official national emissions inventories for many non-Annex I countries. The BURs are also associated with less stringent reporting requirements in terms of sector, gas and time coverage (Deng et al., 2021; Gütschow et al., 2016). This highlights the important role of global inventories such as EDGAR, CEDS, PRIMAP-hist, FAOSTAT or those from IEA or BP among others that are equally as comprehensive in scope to those from Annex I countries. Despite the importance of high-quality emission statistics for climate change research and tracking progress in climate policy, our analysis here emphasises considerable prevailing uncertainties and the need for improvement in emission reporting. Additionally, there are significantly fewer independent estimates for full GHG accounting, in contrast to fossil CO2 emissions. In sectors where production efficiencies are changing rapidly, as is often the case in developing countries, using emission estimates based on Tier-1 methodologies may mischaracterise trends as both activity data and emission factors change over time (Wilkes et al., 2017)."

L981: update Petrescu et al references

Done. Thanks.

L982: non-Annex 1 should be I, and regarding the "lack of recent official GHG emissions inventories" please check the BURs

The BURs are included in the graphical assessment (Figure 9) of the latest submitted national inventories.

L988: complex Tier 3 estimation models are inversions as well (as well as a recommendation of IPCC 2019)

Added to the text. Thanks. See above.

Figure 9 caption: add most recent (2021)

We added the UNFCCC (2021) reference. The latest reporting date for the CRF reports submitted in 2021 is 2019. The map depicts the latest reporting period and not the submission date.

For the updates: even more updated global UNFCCC data

Note sure what is meant by this. We comprehensively also reviewed the most recent submissions of BURs by non-Annex I countries. This map should be fully up-to-date now.

Figure 9: figure could be redone with legend fitting inside and perhaps a different projection

Legend adjusted. Thanks.

Supplementary figures:

Figure SM1: interesting to see the differences between EDGAR versions, but one won't get the feeling on how EDGAR compares to other global estimates, perhaps add UNFCCC to the figures?

We provide the comparison with our dataset (and therefore EDGARv6) in the main manuscript already (Figure 1). We provide additional comparisons with UNFCCC in Figure SM-3 and SM.4. We added a new Figure SM-2 that also provides a comparison for F-gases. Because F-gas data is more limited we did not include them in Figure 1 and therefore plot in this case additional data from US-EPA (2019).

Figure SM3 should state the version of UNFCCC data (2021?) and the link to it.

We added this and highlighted that we access the data via Gütschow et al. (2021).

For all tables some columns should be enlarged to read better the text

Done

**Reviewer 2**

The authors have developed a very comprehensive dataset for GHGs emissions by country, sector, and year. Emissions of $CO_2$, $CH_4$, $N_2O$ and fluorinated gases between 1970 and 2019 are compiled and integrated from a set of commonly used global emission inventories and emission models. Trends and drivers of global emissions have been investigated based on the newly developed emission dataset. Overall, this paper provides useful and interesting results for climate science. I have two concerns about the title and uncertainty assessment of this study, which need to be resolved before it can be accepted for publication in ESSD.

        Thank you for all your efforts and constructive comments. We comprehensively revised the manuscript and hope that this now effectively addresses your concerns.

First, the title of the manuscript seemingly suggests that this study has developed a new and comprehensive global emission dataset. Whilst this is true to a certain extent, it has also to be acknowledged that the new dataset shown here is created only by harmonizing and combining several other inventories and model results, which is not a development effort for a new emission dataset. In my opinion, the title of this work should make it clear that this is an ensemble-based analysis of emission trends and drivers based on existing datasets.

        This is a very good. We changed the title in two ways: A) To address your concern we highlight the synthetic nature of this dataset. While we do compare our dataset with many other datasets as part of the uncertainty assessment, it is not an ensemble analysis as such. It is rather a comprehensive and synthetic dataset with the most recent estimates available to our knowledge that consists of four different components: 1) EDGARv6 data f1970-2018 or $CO_2$-FFI, $CH_4$, $N_2O$ and F-gases; 2) Edgarv6FT fast-track data for $CO_2$-FFI for 2019; 3) $CO_2$-LULUCF data 1970-2019 as the average of three bookkeeping models from the Global Carbon Budget; 4) EDGARv5FT fast-track data for CH4, N2O, F-gas emissions as the basis "for initial 2019 estimates. B) The title now also reflects the new focus on 1970-2018, where we have data from full releases and clarify the preliminary nature of 2019 estimates as well. Please also refer to the overall responses provided by the author team that explain other important changes undertaken in the manuscript. The new title is:

"A comprehensive and synthetic dataset for global, regional and national greenhouse gas emissions by sector 1970-2018 with an extension to 2019"

Second, I do not agree that uncertainties in this dataset are well characterized (line 921 on page 44). Instead, I found that the estimates of uncertainties in this study are rather arbitrary. For example, line 375 on page 18 said that "We base our uncertainty assessment on Friedlingstein et al. (2020) and take $\pm 2.6$ $GtCO_2$ $yr^{-1}$ as a best-value judgment for the $\pm 1\sigma$ uncertainty range (thus $\pm 5.1$ $GtCO_2$ $yr^{-1}$ for $\pm 2\sigma$) in $CO_2$-LULUCF emissions, constant over the last decades.". Line 483 on page 23 said that "Overall, we apply a best value judgment of $\pm 30\%$ for global anthropogenic methane emissions for a 90% confidence interval.". Although I know that estimating emission uncertainties is quite difficult, I did not fully understand how these values of uncertainties were determined in this study. And I do not think that the quantification of uncertainties in this work has been improved compared to previous literature. Besides, I noticed that the methods used to quantify uncertainties are actually different across $CO_2$, $CH_4$, and $N_2O$. Are these uncertainty values comparable to each other in this study.

Thanks for this comment. We agree that our approach to the assessment of uncertainties needed more clarity. We extended the relevant passages in the introduction to the uncertainty section (3) as follows:

"This section provides an assessment of uncertainties in greenhouse gas emissions data at the global level. The uncertainties reported here combine statistical analysis, comparisons of global emissions inventories and expert judgement of the likelihood of results lying outside a defined confidence interval, rooted in an understanding gained from the relevant literature. At times, we also use a qualitative assessment of confidence levels to characterize the annual estimates from each term based on the type, amount, quality, and consistency of the evidence as defined by the IPCC (IPCC, 2014).

Such a comprehensive uncertainty assessment covering all major groups of greenhouse gases and considering multiple lines of evidence has been missing in the literature. The absence has provided a serious challenge for a transparent, scientific reporting of GHG emissions in climate change assessments like those by IPCC's Working Group III or the UN Emissions Gap Report that have only more recently started to even deal with the issue (e.g. Blanco G. et al., 2014; UNEP, 2019). Most of the available studies in the peer-reviewed literature using multiple lines of evidence for their assessment have focussed on individual gases like in the Global Carbon Budget (Friedlingstein et al., 2020), the Global Methane Budget (Saunois et al., 2020) or the Global Nitrous Oxide Budget (Tian et al., 2020) or covered multiple gases, but mainly considered individual lines of evidence (Janssens-Maenhout et al., 2019; Solazzo et al., 2021).

We adopt a 90% confidence interval (5th-95th percentile) to report the uncertainties in our GHG emissions estimates, i.e., there is a 90 % likelihood that the true value will be within the provided range if the errors have a Gaussian distribution, and no bias is assumed. This is in line with previous reporting in IPCC AR5 (Blanco et al., 2014; Ciais et al., 2014). We note that national emissions inventory submissions reported to the UNFCCC are requested to report uncertainty using a 95% or $2\sigma$ confidence interval. The use of this broader uncertainty interval implies, however, a relatively high degree of knowledge about the uncertainty structure of the associated data, particularly regarding the distribution of uncertainty in the tails of the probability distributions. Such a high degree of knowledge is not present over the emission sectors and species considered here. Note that in some cases below we convert $1\sigma$ uncertainty results from the literature to a 90% confidence interval by implicitly assuming a normal distribution. While we do this as a necessary assumption to obtain a consistent estimate across all GHGs, we note that this itself is an assumption that may not be valid. We have made use of the best available information in the literature, but note that much more work on uncertainty quantification remains to be done. Using IPPC uncertainty language, we cannot place high confidence in the robustness of most existing uncertainty estimates."

Reviewer 3:

Many researchers 'watch' this product while applauding the authors for this effort. We (they) all want to help authors produce a definitive product of highest quality.

Thank you so much. Much appreciated. Your comments helped to make this better and more transparent.

The comments that follow, extracted from a longer message but conveyed verbatim, come from one interested reader.

"the authors should defend the coherence of a database where non-LULUCF sectors data and LULUCF data appear to be very differently framed. The first set of data is a bottom up approach based on EDGAR, i.e., activity data and default IPCC EF coefficients—an approach that basically mimics how national GHG inventories are made by countries. The second is based on bookkeeping models from the literature, which are completely different from how countries report. Specifically, they do not consider forest carbon sinks  anthropogenic and thus exclude them. This makes the data very different from those from the countries, contrary to the first set. A corollary or perhaps a pre-requisite to that explanation of coherence would be: what is the scope of the database."

We do not see a fundamental contradiction as outlined here. It is our ambition to provide a comprehensive dataset for GHG emissions with broad coverage –and not a dataset that is most aligned with UNFCCC inventories. Other datasets like PRIMAP-hist (Gütschow et al., 2016) are focused on this. Still, we agree with the reviewer that our manuscript lacked explanation and transparency on this important issue. We had acknowledged that country reporting is different from the scientific definition of land-use emissions in the previous version of our manuscript, citing Grassi et al (2018), but we now substantially expand on this issue in response to the reviewer's comment. We have added the following text to the methods description:

"[…] Since in reality anthropogenic $CO_2$-LULUCF emissions co-occur with natural CO2 fluxes in the terrestrial biosphere, models have to be used to distinguish anthropogenic and natural fluxes (Friedlingstein et al., 2020). $CO_2$-LULUCF as reported here is calculated via a bookkeeping approach, as originally proposed by Houghton et al. (2003), tracking carbon stored in vegetation and soils before and after land-use change. Response curves are derived from the literature and observations to describe the temporal evolution of the decay and regrowth of vegetation and soil carbon pools for different ecosystems and land use transitions, including product pools of different lifetimes. These dynamics distinguish bookkeeping models from the common approach of estimating "committed emissions" (assigning all present and future emissions to the time of the land use change event), which is frequently derived from remotely-sensed land use area or biomass observations (Ramankutty et al., 2007). Most bookkeeping models also represent the long-term degradation of primary forest as lowered standing vegetation and soil carbon stocks in secondary forests, and include forest management practices such as wood harvesting.

The scientific definition of $CO_2$-LULUCF emissions used here differs from the one applied in national greenhouse gas inventories (GHGI) (UNFCCC, 2021) or the inventory data provided by FAO (Tubiello et al., 2021). Concretely, this means that inventory data include natural terrestrial fluxes caused by changes in environmental conditions, e.g., effects of rising atmospheric $CO_2$ ("$CO_2$-fertilization"), climate change, and nitrogen deposition -- sometimes

called "indirect effects" as opposed to the direct anthropogenic effects of land-use change and management (Houghton et al., 2012) - when they occur on area that countries declare as managed. Since environmental changes turned the terrestrial biosphere into a massive sink, removing about one third of annual anthropogenic emissions in the last decade (Friedlingstein et al., 2020), it is unsurprising that global emission estimates are smaller for inventory data than for the scientific definition (see Figure 1). About 3.2 $GtCO2$ $yr^{-1}$ (for the period 2005-2014) was found to be explicable by these conceptual differences in anthropogenic forest sink estimation related to the representation of environmental change impacts and the areas considered as managed (Grassi et al., 2018).

These two conceptually different approaches have different aims: The scientific approach separates natural from anthropogenic drivers, i.e., effects of changes in environmental conditions from effects of land-use change and land management. By contrast, the inventory approach separates fluxes based on areas, with all those occurring on managed land being declared anthropogenic. Given that observational data of carbon stocks or fluxes cannot distinguish the co-occurring effects of environmental changes and land-use activities, an area-based approach that does not require this distinction can more consistently be implemented across countries. These conceptual differences between scientific and inventory approaches have been acknowledged (Canadell et al., 2021; Petrescu et al., 2020a) and approaches have been developed to map the scientific and inventory definitions to each other (Grassi et al., 2018, 2021). For non-$CO_2$ GHGs, drivers and areas coincide, such that FAOSTAT data for $CH_4$ and $N_2O$ is complementary to bookkeeping CO2-LULUCF emissions.

Following the approach taken by the global carbon budget (Friedlingstein et al., 2020), we take the average of three bookkeeping estimates […]"

"... the authors should greatly improve their results and discussion sections. When presenting and comparing data for agriculture ad LULUCF data to existing databases – they should explicitly include a discussion on comparisons to FAO data –after all they have FAOSTAT in Tab. 1, listed among the relevant databases."

Rather than improving only the discussion we made sure that FAOSTAT data is presented more consistently in the manuscript. Most importantly, we added $CO_2$-LULUCF data from FAOSTAT to Figure 1, but also N2O emissions, which were missing previously. We also discuss those estimates in the respective sections. We further improved the discussion in general, but also specifically with regard to the issue of CO2-LULUCF emissions. In that regard, we added the following language:

"We have used a scientific definition for $CO_2$-LULUCF emissions, in line with our intention to identify GHG fluxes attributable to human activities and in line with the GCP's (Friedlingstein et al., 2020) and IPCC WG1's (Canadell et al., 2021) approach that split natural from anthropogenic drivers. We acknowledge that this differs from national GHGI or inventory data provided by FAO (Tubiello et al., 2021), which should be used if consistency in definition with, e.g., the UNFCCC process is sought. Net $CO_2$-LULUCF emissions estimates are substantially smaller based on inventory data over managed areas, because the natural terrestrial sink on managed land is attributed to anthropogenic emissions in this case. It highlights the additional relevance of vegetation in emissions reduction efforts: On the one hand, net emissions from land-use activities should be minimized by reducing gross emissions (e.g., by stopping deforestation and degradation) and increasing gross removals (e.g., by reforestation) (Roe et al., 2019). On the other hand, vegetation acting as natural sink to

anthropogenic $CO_2$ emissions should be kept up, be it managed forest land, as in the inventories, or pristine vegetated lands.

While the distinction between scientific approach (drivers) using bookkeeping models and inventory approach (areas) is clear and mapping approaches have been suggested (Grassi et al., 2018, 2021), the attribution of synergistic terms of environmental and anthropogenic changes differs between methods. Obermeier et al. (2021) showed a strong dependence of $CO_2$-LULUCF emissions estimates on the time a certain land-use change event happened to occur, if environmental changes are represented transiently over time, as is the case for typical simulations with dynamic global vegetation models. This dependence is eliminated by using bookkeeping estimates, as done here.

It should also be mentioned that system boundaries partly differ across datasets, and FAOSTAT data (Tubiello et al., 2021) is currently limited to $CO_2$ fluxes related to forests and emissions from drainage of organic soils under cropland/grassland, excluding other managed land or agricultural conversions. In principal, bookkeeping and DGV models include all fluxes, but are often coarse in their description of management, which observation-based approaches capture."

"... they should acknowledge the existence of a very vibrant discussion (eg grassi et al 2021 nature CC and previous work) where people are arguing for and against this or that approach to estimate LULUCF data, precisely pitting models and bottom up methods ''against each other''.  Which side of the discussion does this work fall on ? what so the comparisons of their data to both approaches say? This is entirely missing from the paper but in my view should be part of their argumentations justifying one more database on GHG emissions – especially the LULUCF part."

Please note that we had referenced Grassi et al. (2018) (of which Grassi et al., 2021, is a methodological update), but framed it under uncertainties, so maybe it has been missed by the reviewer -- we fully agreed it is an important aspect. As can be seen from the added text we extended this discussion now substantially both in the methods section as well as in the discussion section to make this aspect more prominent. Please note that we do not support a "pitting [approaches] against each other", as the different approaches simply have different aims. This is also the way how it is depicted in the Grassi et al papers. We hope we could clarify this with our text additions. We thank the reviewer for all comments, which helped make this a better manuscript.